



# The consolidated European synthesis of $CO_2$ emissions and removals for EU27 and UK: 1990-2018

Ana Maria Roxana Petrescu[1], Matthew J. McGrath[2], Robbie M. Andrew[3], Philippe Peylin[2], Glen P. Peters[3], Philippe
Ciais[2], Gregoire Broquet[2], Francesco N. Tubiello[4], Christoph Gerbig[5], Julia Pongratz[6,7], Greet Janssens-Maenhout[8],
Giacomo Grassi[8], Gert-Jan Nabuurs[9], Pierre Regnier[10], Ronny Lauerwald[10,11], Matthias Kuhnert[12], Juraj Balkovič[13,14],
Mart-Jan Schelhaas[9], Hugo A. C. Denier van der Gon[15], Efisio Solazzo[8], Chunjing Qiu[2], Roberto Pilli[8], Igor B.
Konovalov[16], Richard Houghton[17], Dirk Günther[18], Lucia Perugini[19], Monica Crippa[9], Raphael Ganzenmüller[6], Ingrid
T. Luijkx[9], Pete Smith[12], Saqr Munassar[5], Rona L. Thompson[20], Giulia Conchedda[4], Guillaume Monteil[21], Marko
Scholze[21], Ute Karstens[22], Patrick Brokmann[2] and Han Dolman[1]

[1]Department of Earth Sciences, Vrije Universiteit Amsterdam, 1081HV, Amsterdam, the Netherlands
[2]Laboratoire des Sciences du Climat et de l'Environnement, CEA CNRS UVSQ UPSACLAY Orme des Merisiers,
Gif-sur-Yvette, France
[3]CICERO Center for International Climate Research, Oslo, Norway
[4]FAO, Statistics Division, Via Terme di Caracalla, Rome 00153, Italy
[5]Max Planck Institute for Biogeochemistry, Hans-Knöll-Strasse 10, 07745 Jena, Germany
[6]Department of Geography, Ludwig Maximilian University of Munich, 80333 Munich, Germany
[7]Max Planck Institute for Meteorology, Bundesstrasse 53, 20146 Hamburg, Germany
[8]European Commission, Joint Research Centre, Via Fermi 2749, 21027 Ispra, Italy
[9] Wageningen Environmental Research, Wageningen University and Research (WUR), Wageningen, 6708PB, the
Netherlands
[10] Biogeochemistry and Modeling of the Earth System, Université Libre de Bruxelles, 1050 Brussels, Belgium
[11]Université Paris-Saclay, INRAE, AgroParisTech, UMR ECOSYS, Thiverval-Grignon, France
[12]Institute of Biological and Environmental Sciences, University of Aberdeen, 23 St Machar Drive, Aberdeen, AB24
3UU, UK
[13]International Institute for Applied Systems Analysis, Ecosystems Services and Management Program, Schlossplatz
1, A-2361, Laxenburg, Austria
[14]Faculty of Natural Sciences, Comenius University in Bratislava, Ilkovičova 6, 842 15, Bratislava, Slovak Republic
[15]TNO, Department of Climate, Air and Sustainability, Princetonlaan 6, 3584 CB Utrecht, the Netherlands
[16]Institute of Applied Physics, Russian Academy of Sciences, Nizhniy Novgorod, Russia
[17]Woods Hole Research Center, Falmouth, Massachusetts, U.S.A.
[18]Umweltbundesamt (UBA), 14193 Berlin, Germany
[19]Centro Euro-Mediterraneo sui Cambiamenti Climatici (CMCC), Viterbo, Italy
[20]Norwegian Institute for Air Research (NILU), Kjeller, Norway
[21]Dept. of Physical Geography and Ecosystem Science, Lund University, Lund, Sweden
[22]ICOS Carbon Portal at Lund University, Lund, Sweden

*Correspondence to*: A.M. Roxana Petrescu (a.m.r.petrescu@vu.nl)

**Abstract**

Reliable quantification of the sources and sinks of atmospheric carbon dioxide ($CO_2$), including that of their
trends and uncertainties, is essential to monitoring the progress in mitigating anthropogenic emissions under the Kyoto
Protocol and the Paris Agreement. This study provides a consolidated synthesis of estimates for all anthropogenic and
natural sources and sinks of $CO_2$ for the European Union and UK (EU27 + UK), derived from a combination of state-
of-the-art bottom-up (BU) and top-down (TD) data sources and models. Given the wide scope of the work and the
variety of datasets involved, this study focuses on identifying essential questions which need to be answered to
properly understand the differences between various datasets, in particular with regards to the less-well characterized



fluxes from managed ecosystems. The work integrates recent emission inventory data, process-based ecosystem
model results, data-driven sector model results, and inverse modelling estimates, over the period 1990-2018. BU and
TD products are compared with European national GHG inventories (NGHGI) reported under the UNFCCC in 2019,
aiming to assess and understand the differences between approaches. For the uncertainties in NGHGI, we used the
standard deviation obtained by varying parameters of inventory calculations, reported by the Member States following
the IPCC guidelines. Variation in estimates produced with other methods, like atmospheric inversion models (TD) or
spatially disaggregated inventory datasets (BU), arise from diverse sources including within-model uncertainty related
to parameterization as well as structural differences between models. In comparing NGHGI with other approaches, a
key source of uncertainty is that related to different system boundaries and emission categories ($CO_2$ fossil) and the
use of different land use definitions for reporting emissions from Land Use, Land Use Change and Forestry (LULUCF)
activities ($CO_2$ land). At the EU27+UK level, the NGHGI (2019) fossil $CO_2$ emissions (including cement production)
account for 2624 Tg $CO_2$ in 2014 while all the other seven bottom-up sources are consistent with the NGHGI and
report a mean of 2588 ($\pm$ 463 Tg $CO_2$). The inversion reports 2700 Tg $CO_2$ ($\pm$ 480 Tg $CO_2$), well in line with the
national inventories. Over 2011-2015, the $CO_2$ land sources/sinks from NGHGI estimates report -90 Tg C yr$^{-1}$ $\pm$ 30
Tg C while all other BU approaches report a mean sink of -98 Tg yr$^{-1}$ ($\pm$ 362 Tg C from DGVMs only). For the TD
model ensemble results, we observe a much larger spread for regional inversions (i.e., mean of 253 Tg C yr$^{-1}$ $\pm$ 400
Tg C yr$^{-1}$). This concludes that a) current independent approaches are consistent with NGHGI b) their uncertainty is
too large to allow a "verification" because of model differences and probably also because of the definition of "$CO_2$
flux" obtained from different approaches. The referenced datasets related to figures are visualized at
https://doi.org/10.5281/zenodo.4288883 (Petrescu et al., 2020).

## 1. Introduction

Global atmospheric concentrations of $CO_2$ have increased 46% since pre-industrial times (pre-1750) (WMO,
2019). The rise of $CO_2$ concentrations in recent decades is caused primarily by $CO_2$ emissions from fossil sources.
Globally, fossil emissions grew at a rate of 1.3% yr$^{-1}$ for the decade 2009–2018 and accounted for 87% of the
anthropogenic sources in the total carbon budget (Friedlingstein et al., 2019). In contrast, global $CO_2$ emissions from
land use and land use change estimated from bookkeeping models and dynamic global vegetation models (DGVMs)
were approximately stable during the same period, albeit with large uncertainties (Friedlingstein et al., 2019).

National GHG inventories (NGHGI) are prepared and reported under the UNFCCC on an annual basis by
Annex I countries[1], based on IPCC Guidelines using national activity data and different levels of sophistication (tiers)
for well-defined sectors. These inventories contain time series of annual GHG emissions from the 1990 base year[2]

---

[1] Annex I Parties include the industrialized countries that were members of the OECD (Organization for Economic Co-operation and Development)
in 1992 plus countries with economies in transition (the EIT Parties), including the Russian Federation, the Baltic States, and several central and
eastern European states (UNFCCC, https://unfccc.int/parties-observers, last access: February 2020).

[2] For most Annex I Parties, the historical base year is 1990. However, parties included in Annex I with an economy in transition during the early
1990s (EIT Parties) were allowed to choose one year up to a few years before 1990 as reference because of a non-representative collapse during the
breakup of the Soviet Union (e.g., Bulgaria, 1988, Hungary, 1985–1987, Poland, 1988, Romania, 1989, and Slovenia, 1986).



until two years before the current year and were required by the UNFCCC and used to track progress towards countries' reduction targets under the Kyoto Protocol (UNFCCC, 1997). The IPCC tiers represent the level of sophistication used to estimate emissions, with Tier 1 based on global or regional default values, Tier 2 based on country and technology-specific parameters, and Tier 3 based on more detailed process-level modelling. Uncertainties in NGHGI are calculated based on ranges in observed (or estimated) emission factors and variation of activity data,

using the error propagation method (95% confidence interval) or Monte-Carlo methods, based on clear guidelines (IPCC 2006).

        NGHGIs follow principles of transparency, accuracy, consistency, completeness and comparability (TACCC) under the guidance of the UNFCCC (UNFCCC, 2014). Methodological procedures follow the 2006 IPCC guidelines (IPCC, 2006) and can be upgraded and completed with the IPCC 2019 Refinement (IPCC, 2019a) containing updated sectors and additional sources. Atmospheric GHG concentration data can be used to derive

estimates of the GHG fluxes based on atmospheric transport inverse modeling techniques (Rayner et al., 2019). Such estimates are often called top-down (TD) estimates since these are based on the analysis of concentrations, which represent sum of the effect of source and sinks, in contrast to bottom-up (BU) estimates, which rely on models analyzing the processes causing the fluxes. Current UNFCCC procedures do not require observation-based evidence

in the NGHGI and do not incorporate independent, large-scale observation-based GHG budgets but the latest guidelines allow the use of atmospheric data for external checks within the data quality control, quality assurance and verification process (IPCC 2006 Guidelines, Chapter 6 QA/QC procedures). Only a few countries (e.g. Switzerland, UK, New Zealand and Australia) use atmospheric observations on a voluntary basis to complement their national inventory data with top down estimates annexed to their NGHGI (Bergamaschi et al., 2018).

For the post-2020 reporting (which will start in 2023 for the inventory of year 2021), the Paris Agreement follows on the Kyoto Protocol and, at the EU level, the GHG Monitoring Mechanism Regulation 525 (2013) is replaced by Regulation 1999 (2018) while Regulation 824 (2018) embeds the LULUCF sector with estimates based on spatial information in the EU Climate Targets of 2030. A key element in the current policy process is to facilitate the global stocktake exercise of the UNFCCC foreseen in 2023, which will assess collective progress towards

achieving the near- and long-term objectives of the Paris Agreement, also considering mitigation, adaptation and means of implementation. The global stocktake is expected to create political momentum for enhancing commitments in Nationally Determined Contributions (NDCs) under the Paris Agreement.

        Key components of the global stocktake are the NGHGI submitted by countries under the enhanced transparency framework of the Paris Agreement. Under the new framework, for the first time, developing countries

will be required to submit their inventories on a biennial basis, alongside developed countries that will continue to submit their inventories and full time-series on an annual basis. This calls for robust and transparent approaches that can build-up long-term emission compilation capabilities and be applied to different situations. A priority is to refine estimates of $CH_4$ and $N_2O$ emissions, which are more uncertain than the $CO_2$ fossil emissions. Fossil $CO_2$ emissions are closely anchored to well established fuel use statistics with narrow uncertainty ranges on emissions factors, while

$CO_2$ from LULUCF and $CH_4$ and $N_2O$ have highly uncertain activity data and/or emission factors (see companion



paper, Petrescu et al., submitted). However, $CO_2$ emissions dominate the GHG fluxes and there is need for Monitoring and Verification Support capacity (Janssen-Meanhout et al., 2020) as the reduction of anthropogenic $CO_2$ fluxes become increasingly important for the climate negotiations of the Paris Agreement, and where observation-based data can provide information on the actual situation. In addition, while fossil $CO_2$ emissions are known to relatively high precision, LULUCF activities are generally much more uncertain (RECCAP, CarboEurope) and as described below in sections 2.2. and 3.2.

The current study presents consistently derived estimates of $CO_2$ fluxes from BU and TD approaches for the EU27 and UK, building partly on Petrescu et al. (2020) for the LULUCF sector and on Andrew (2020) for fossil sectors, while laying the foundation for future annual updates. Every year (time "t") the Global Carbon Project (GCP) in its Global Carbon Budget (GCB) quantifies large-scale $CO_2$ budgets up to year "t-1", bringing in information from global to large latitude bands, including various observation-based flux estimates from BU and TD approaches (Friedlingstein et al., 2020 in review). Except for two sector-specific BU models based on national statistics (EFISCEN and CBM), we note that the BU observation-based approaches used in the GCB and in this paper are based on the NGHG estimates provided by national inventory agencies to the UNFCCC with differences coming from allocation. They rely heavily on statistical data combined with Tier1 and Tier2 approaches. In our case, focusing on a region that is well covered with data and models (Europe), BU also refers to Tier 3 process-based models or complex bookkeeping models (see section 2). At regional and country scales, no systematic and regular comparison of these observation-based $CO_2$ flux estimates with reported fluxes at UNFCCC is yet feasible. As a first step in this direction, within the European project VERIFY (http://verify.lsce.ipsl.fr/), the current study compares observation-based flux estimates of BU versus TD approaches and compares them with NGHGIs for the EU27+UK and five sub-regions (Figure 4). The methodological and scientific challenges to compare these different estimates have been partly investigated before (Grassi et al., 2018 for LULUCF; Peters et al., 2009 for fossil sectors) but not in a systematic and comprehensive way including both fossil and land-based $CO_2$ fluxes.

The work presented here represents many distinct datasets and use of models in addition to the individual country submissions to the UNFCCC for all European countries, which while following the general guidance laid out in IPCC (2006) still differ in specific approaches, models, and parameters, in addition to differences in underlying activity datasets. A comprehensive investigation of detailed differences between all datasets is beyond the scope of this paper, though attempts have been previously made for specific subsectors (Petrescu et al., 2020 for AFOLU[3]; Federici et al., 2015 for FAOSTAT versus NGHGIs). As this is the most comprehensive comparison of NGHGIs and research datasets (including both bottom-up (BU) and top-down (TD) approaches) for Europe to date, we focus here on a set of questions that such a comparison raises: How can one fairly compare the detailed sectoral NGHGI to observation-based estimates? What new information do the observation-based estimates provide, for instance on the mean fluxes, spatial disaggregation, trends and inter-annual variation? What can one expect from such complex studies, where are the key knowledge gaps, what is the added value to policy makers and what are the next steps to take?

---

[3] In the IPCC AR5 AFOLU stands for Agriculture, Forestry and Other Land Use and represent a new sector replacing the two AR4 sectors Agriculture and LULUCF





We compare official anthropogenic NGHGI emissions with research datasets correcting wherever needed research data on total emissions/sinks to separate out anthropogenic emissions. We analyze differences and inconsistencies between emissions and sinks, and make recommendations towards future actions to evaluate NGHGI data. While NGHGI include uncertainty estimates, special disaggregated research datasets of emissions often lack
quantification of uncertainty. While this is also a call to those developers to associate more detailed uncertainty estimates with their products, here we use the median and minimum/maximum (min/max) range of different research products of the same type to get a first estimate of overall uncertainty. Table AA in Appendix A presents the methodological differences of current study with respect to Petrescu et al., 2020.

**2. CO₂ data sources and estimation approaches**

We use data of total $CO_2$ emissions and removals from EU27 + UK from TD inversions and BU estimates, in addition to BU estimates from sector-specific models. We collected data of $CO_2$ fossil and $CO_2$ land[4] emissions and removals between 1990 and 2018 (or the last available year, if the datasets do not extend to 2018) from peer-reviewed literature and other data delivered under the VERIFY project (see description in Appendix A). The detailed data source
descriptions are found in Appendices A1 and A2. For the BU anthropogenic $CO_2$ fossil estimates we used global inventory datasets (EDGAR v5.0.0, FAOSTAT, BP, CDIAC, GCP, EIA, IEA, see Table 1) described in detail by Andrew (2020), while for $CO_2$ land estimates we used BU research-level biogeochemical models (e.g. DGVMs TRENDY-GCP, bookkeeping models, see Table 2). For TD we used global inversions (GCP 2019, Friedlingstein et al., 2019) as well as regional inversions at higher spatial resolution (CarboScopeReg, EUROCOM (Monteil et al.,
2019 and Konovalov et al. 2016).

The values are defined from an atmospheric perspective: positive values represent a source to the atmosphere and negative ones a removal from the atmosphere. As an overview of potential uncertainty sources, Appendix B presents the use of emission factor data (EF), activity data (AD), and, whenever available, uncertainty methods used for all $CO_2$ land data sources used in this study. The referenced data used for the figures' replicability purposes are
available for download at https://doi.org/10.5281/zenodo.4288883 (Petrescu et al., 2020). We focus herein on EU27 and the UK. Within the VERIFY project, we have in addition constructed a web tool which allows for the selection and display of all plots show in this paper (as well as the companion paper on CH₄ and N₂O), not only for the regions shown here but for a total of 79 countries and groups of countries in Europe. The website, located on the VERIFY project website: http://webportals.ipsl.jussieu.fr/VERIFY/FactSheets/, is accessible with a username and password
distributed by the project. Figure 4 includes also data from countries outside the EU but located within geographical Europe (Switzerland, Norway, Belarus, Ukraine and Rep. of Moldova).

---

[4] The IPCC Good Practice Guidance (GPG) for Land Use, Land Use Change and Forestry (IPCC 2003) describes a uniform structure for reporting emissions and removals of greenhouse gases. This format for reporting can be seen as "land based"; all land in the country must be identified as having remained in one of six classes since a previous survey, or as having changed to a different (identified) class in that period. According to IPCC SRCCL: Land covers the terrestrial portion of the biosphere that comprises the natural resources (soil, near surface air, vegetation and other biota, and water) the ecological processes, topography, and human settlements and infrastructure that operate within that system".



### 2.1. CO₂ anthropogenic emissions from NGHGI

UNFCCC NGHGI (2019) emissions are country estimates covering the period 1990-2017. The Annex-I parties to the UNFCCC are required to report emissions inventories annually using the Common Reporting Format (CRF). This annual published dataset includes all $CO_2$ emissions sources for those countries, and for most countries for the period 1990 to t-2. Some eastern European countries' submissions begin in the 1980s. Revisions are made on an irregular basis outside of the standard annual schedule.

### 2.2. CO₂ fossil emissions

$CO_2$ fossil emissions occur when fossil carbon compounds are broken down via combustion or other forms of oxidation or via non-metal processes such as for cement production. Most of these fossil compounds are in the form of fossil fuels, such as coal, oil, and natural gas. Another category are fossil carbonates, such as calcium carbonate and magnesium carbonate, which are used as feed stocks in industrial processes, and whose decomposition also leads

to emissions of $CO_2$. Because $CO_2$ fossil emissions are largely connected with energy, which is a closely tracked commodity group, there is a wealth of underlying data that can be used for estimating emissions. However, differences in collection, treatment, interpretation and inclusion of various factors such as carbon contents and fractions of oxidized carbon, lead to methodological differences (Appendix A, Table A1) resulting in differences of emissions between datasets (Andrew 2020). In contrast to BU estimates, atmospheric inversions for emissions of fossil $CO_2$ are

not fully established (Brophy et al., 2019), though estimates exist. The main reason is that the types of atmospheric networks suitable for fossil $CO_2$ atmospheric inversions have not been widely deployed yet (Ciais et al. 2015).

In this analysis, the BU $CO_2$ fossil estimates are presented and split per fuel type and reported for the last year when all data products are available (Andrew 2020). In addition to the BU $CO_2$ fossil estimates, we report a fossil fuel $CO_2$ emission estimate for the year 2014 from a 4-year inversion assimilating satellite observations. In order to

overcome the lack of $CO_2$ observation networks suitable for the monitoring of fossil fuel $CO_2$ emissions at national scale, this inversion is based on atmospheric concentrations of co-emitted species. It assimilates satellite CO and $NO_2$ data. While the spatial and temporal coverage of these CO and $NO_2$ observations is large, the conversion of the information on these co-emitted species into fossil fuel $CO_2$ emission estimates is complex and carries large uncertainties. Therefore, we focus here on the comparison between the uncertainties in the inversion versus the

magnitude and variations of BU estimates without discussing system boundaries and constraints of each of these products (which are instead discussed in Andrew 2020). The detailed descriptions of each of the data products described in Table 1 are found in Appendix A1.

*Table 1: Data sources for the anthropogenic CO₂ fossil emissions included in this study:*

| CO₂ anthropogenic | | | |
|---|---|---|---|
| **Data/model name** | **Contact / lab** | **Species / Period** | **Reference/Metadata** |
| UNFCCC NGHGI (2019) | UNFCCC | Anthropogenic fossil $CO_2$ 1990-2017 | IPCC, 2006 IPCC Guidelines for National Greenhouse Gas Inventories |

| | | | | https://www.ipcc-nggip.iges.or.jp/public/2006gl/, 2006. <br><br> UNFCCC CRFs https://unfccc.int/process-and-meetings/transparency-and-reporting/reporting-and-review-under-the-convention/greenhouse-gas-inventories-annex-i-parties/national-inventory-submissions-2019 |
|---|---|---|---|---|
| BU | Compilation of multiple $CO_2$ fossil emission data sources (Andrew 2020) EDGAR v5.0, BP, EIA, CDIAC, IEA, GCP, CEDS, PRIMAP | CICERO | $CO_2$ fossil country totals and split by fuel type 1990-2018 (or last available year) | EDGAR v5.0 https://edgar.jrc.ec.europa.eu/overview.php?v=50_GHG BP 2011, 2017 and 2018 reports EIA https://www.eia.gov/beta/international/data/browser/views/partials/sources.html CDIAC https://energy.appstate.edu/CDIAC https://www.eia.gov/beta/international/data/browser/views/partials/sources.html IEA https://www.transparency-partnership.net/sites/default/files/u2620/the_iea_energy_data_collection_and_co2_estimates_an_overview__iea__coent.pdf. IEA, 2018d, p. I.17 CEDS http://www.globalchange.umd.edu/data-products/ GCP Le Quéré et al., 2018, Friedlingstein et al., 2019 https://www.icos-cp.eu/GCP/2018 PRIMAP https://dataservices.gfz-potsdam.de/pik/showshort.php?id=escidoc:2959897 |
| TD | Fossil fuel $CO_2$ inversions | IAPRAS | Inverse fossil fuel $CO_2$ emissions 2012-2015 | Konovalov et al., 2016 VERIFY report https://projectsworkspace.eu/sites/VERIFY/WP%20documents/Estimate-FFCO2-Europe-2012-2015-Konovalov-et-al.pdf |

## 215   2.3.  $CO_2$ land fluxes

$CO_2$ land fluxes include $CO_2$ emissions and removals from LULUCF activities, based on either BU or TD $CO_2$ estimates from inversion ensembles, represented by the data sources and products described in Table 2. We compare $CO_2$ net emissions from the LULUCF sector primarily from three land use classes[5] (Forest Land, Cropland, and Grassland) from both land class remaining[6] (land class remains unchanged) and land class converted[7] (land class changed in the last 20 years). The Wetlands, Settlements, Other Land categories are included in the discussion on total LULUCF activities (incl. Harvested Wood Products (HWP)) presented in sections 3.3.1, 3.3.3 and 3.3.4. Not all the

---

[5] According to 2006 IPCC guidelines the LULUCF sector includes six management classes (forest land, cropland, grassland, wetlands, settlements and other land)

[6] According to 2006 IPCC guidelines, land should be reported in a "conversion" category for 20 years and then moved to a "remaining" category, unless a further change occurs. Converted land refers to $CO_2$ emissions from conversions to and from all six classes that occurred in the previous 20 years.

[7] Converted land refers to $CO_2$ emissions from conversions to and from all six classes that occurred in the previous 20 years.

classes reported to the UNFCCC are present in FAOSTAT or other models; in addition some models are sector-specific. We use the notation of "FL-FL", "CL-CL", and "GL-GL" to indicate forest, cropland, and grassland which

remain the same class from year to year. We present separate results from sector-specific models reporting carbon fluxes for FL-FL, CL-CL and GL-GL (the models EPIC-IIASA, ECOSSE, EFISCEN, CBM), those including multiple land use sectors and simulating land use changes ( e.g. Dynamic Global Vegetation Models (DGVMs) ensemble TRENDY v7 (Sitch et al., 2008, Le Quéré et al., 2009)), and those employing bookkeeping approaches (H&N (Houghton & Nassikas, 2017) and BLUE (Hansis et al., 2015)). The detailed description of each of the products

described in Table 3 is found in Appendix A2.

The two inverse model ensembles presented here are the GCB 2018 for 1990-2018 (Le Queré et al., 2018) and EUROCOM for 2006-2015 (Monteil et al., 2019). The GCB inversions are global and include CarbonTracker Europe (CTE; van der Laan-Luijkx et al., 2017), CAMS (Chevallier et al., 2005) and the Jena CarboScope (Rödenbeck, 2005). The EUROCOM inversions are regional, with a domain limited to Europe and higher spatial

resolution atmospheric transport modes, with five inversions covering the entire period 2006-2015 as analyzed in Monteil et al. (2019). They report Net Ecosystem Exchange (NEE) fluxes. These inversions make use of more than 30 atmospheric observing stations within Europe, including flask data and continuous observations and work at typically higher spatial resolution then the global inversion models. The other regional inversion presented here is generated with the CarboScope-Regional (CSR) inversion system (2006-2018), with different ensemble members.

This system is part of the EUROCOM ensemble, but new runs were carried out for the VERIFY project. The results are plotted separately to illustrate two points: 1) that the CSR runs for VERIFY are not identical to those submitted to EUROCOM (VERIFY runs from CSR included several sites that started shortly before the end of the EURCOM inversion period), and 2) the CSR model was used in four distinct runs in VERIFY, that differing in the spatial correlation of prior uncertainties and in the number of atmospheric stations whose observations are assimilated. By

presenting CSR separate from the EUROCOM results, one can get an idea of the uncertainty due to various model parameters in one inversion system, with one single transport model.


*Table 2: Data sources for the land CO$_2$ emissions included in this study:*

| Bottom-up NGHGI CO$_2$ land | | | | |
|---|---|---|---|---|
| Met hod | Product Type / file or directory name | Contact / lab | Variables / Period | References |
| | **UNFCCC NGHGI (2019)** | UNFCCC | LULUCF Net CO$_2$ emissions/removals 1990-2017 | IPCC, 2006 Guidelines for National Greenhouse Gas Inventories, Prepared by the National Greenhouse Gas Inventories Programme. IGES, Japan, https://www.ipcc-nggip.iges.or.jp/public/2006gl/, 2006. |



| | | | | UNFCCC CRFs<br><br>https://unfccc.int/process-and-meetings/transparency-and-reporting/reporting-and-review-under-the-convention/greenhouse-gas-inventories-annex-i-parties/national-inventory-submissions-2019 |
|---|---|---|---|---|
| | | | **Observation-based bottom-up CO₂ land** | |
| BU | **ORCHIDEE** | LSCE | CO₂ fluxes and C stocks from forest, cropland, and grassland ecosystems reported as Net Biome Productivity (NBP) 1990-2018 | Ducoudré et al., 1993<br>Viovy et al., 1996<br>Polcher et al., 1998<br>Krinner et al., 2005 |
| BU | **CO₂ emissions from inland waters** | ULB | One average value for C fluxes from rivers, lakes and reservoirs, with lateral C transfer from soils<br><br>1990-2018 | Lauerwald et al. (2015)<br>Hastie et al. (2019)<br>Raymond et al. (2013) |
| BU | **CBM** | EC-JRC | Net primary production (NPP) and carbon stocks and fluxes 2000-2015 | Kurz et al., 2009<br>Pilli et al., 2016b |
| BU | **ECOSSE grasslands, croplands** | UNIABDN | CO₂ fluxes from croplands and grassland ecosystems, with a particular focus on soils / Rh, NEE and NBP 1990-2018 | Bradbury et al., 1993<br>Coleman., 1996<br>Jenkinson., 1977, 1987<br>Smith et al., 1996, 2010a,b |
| BU | **EFISCEN** | WUR | Forest biomass and soils C stocks and NBP (a single average value for 5 year periods, replicated on a yearly time axis) | Verkerk et al., 2016<br>Schelhaas et al. 2017<br>Nabuurs et al., 2018 |
| BU | **EPIC-IIASA croplands** | IIASA | CO₂ emissions from cropland 1981-2018 | Balkovič et al., 2013, 2018<br>Izaurralde et al., 2006<br><br>Williams et al., 1990 |
| BU | **BLUE bookkeeping model for land use change** | MPI/LMU Munich | Net C flux from land use change, split into the contributions of different types of land use (cropland vs pasture expansion, afforestation, wood harvest) 1970-2017 | Hansis et al., 2015<br>Le Quéré et al., 2018 |
| BU | **H&N bookkeeping model** | | C flux from land use and land cover 1990-2015 | Houghton & Nassikas, 2017 |
| BU | **FAO** | FAOSTAT | CO₂ emissions/removal from LULUCF sectors 1990-2017 | FAO, 2014<br>Federici et al., 2015<br>Tubiello, 2019 |



| BU | **TRENDY v7 (2018) models: : CABLE, CLASS, CLM5, DLEM, ISAM, JSBACH, JULES, LPJ, LPX, OCN, ORCHIDEE-CNP, ORCHIDEE, SDGVM, SURFEX** | MetOffice UK | Land related C emissions (NBP) from 14 bottom up models 1900-2017 | References for all models in Le Quéré et al., 2018 https://www.icos-cp.eu/GCP/2018 |
|---|---|---|---|---|
| | **Top-down CO₂ estimates** | | | |
| TD | **CarboScope regional inversions** | MPI -Jena | Total CO₂ inverse flux 2006-2018 | Kountouris et al 2018 a,b |
| TD | **GCB 2019 Global inversions (CTE, CAMS, CarboScope)** | GCP | Total CO₂ inverse flux (NBP) 4 inversions 1985-2018 | Friedlingstein et al., 2019 Van der Laan-Luijk et al., 2017 Chevallier et al., 2005 Rödenbeck et al., 2005 |
| TD | **EUROCOM regional inversions 2019, 7 inversions (incl. CarboScope-Reg)** | LSCE | Total CO₂ inverse flux (NBP) 2006-2015 2006-2018 (CarboScopeReg) | Monteil et al., 2019 |

## 3. Results and discussion

### 3.1. Overall NGHGI reported fluxes

According to UNFCCC NGHGI (2019) estimates, in 2017 the European Union (EU27 + UK) emitted 3.96 Gt $CO_2$eq from all sectors (incl. LULUCF) and 4.21 Gt $CO_2$eq (excl. LULUCF) (Appendix B1, Figure B1a). LULUCF only contributed 0.28 Gt $CO_2$ in 2017. This number is consistent with a variety of independent emission inventories (Andrew 2020 and Petrescu et al., 2020). A few large economies account for the largest share of EU27 + UK emissions, with Germany, UK and France representing 43 % of the total $CO_2$ emissions (excl. LULUCF) in 2017. For

LULUCF the countries reporting the largest $CO_2$ sinks were Sweden, Poland and Spain accounting for 45% of the overall EU27 + UK sink strength. Only a few countries (The Netherlands, Ireland, Portugal and Denmark) reported a net LULUCF source in 2017; in the case of Portugal, this was mainly due to emissions from biomass burning. The UNFCCC show minimal inter-annual variability, so the 2017 values are indicative of longer-term trends.

        $CO_2$ fossil emissions are dominated by the energy sector, combustion and fugitives, representing 91.4 % of

the total EU27 + UK $CO_2$ emissions (excl. LULUCF) or 3.25 Gt $CO_2$ yr$^{-1}$ in 2017. The Industrial Process and Product Use sector (IPPU) sector contributes 8.2 % or 0.2 Gt $CO_2$ yr$^{-1}$ while the $CO_2$ emissions reported as part of the



agriculture sector cover only liming and urea application, UNFCCC sectors 3G and 3H[8] respectively. Together with waste, in 2017, the emissions from agriculture represent 0.4 % of the total UNFCCC $CO_2$ emissions. Often, the NGHGI reported values for $CO_2$ emissions do not include LULUCF as these reported-emissions are inherently

uncertain showing almost no inter-annual variability, contrary to observation-based BU approaches (e.g. process-based models) which do show large inter-annual variations as a result of inter-annual variability in climatic conditions, and (in part as a consequence of this variability) in the occurrence of natural disturbances (Kurz et al., 2010, Olivier et al., 2017).

**3.2. $CO_2$ fossil emissions**

*Bottom-up estimates by sector*

At the EU27+UK level our results show that $CO_2$ fossil emissions are consistent between UNFCCC NGHGI (2019) and BU inventories from EDGAR v5.0, CEDS, and PRIMAP. EDGAR v5.0 reports the same sources as the UNFCCC, but CEDS reports emissions from Energy (1A+1B), IPPU and Waste up to 2014, and PRIMAP only for

Energy and IPPU. All BU datasets show a good match for overlapping sectors, Energy and IPPU (Fig. 1, sum of sub-sectors 1A, 1B).

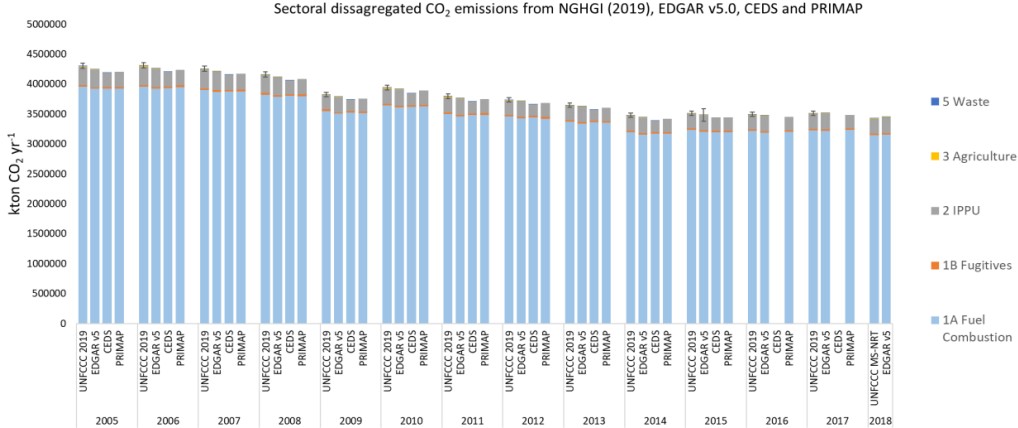

*Figure 1: Total sectoral breakdown of $CO_2$ fossil emissions from UNFCCC NGHGI (2019), EDGAR v5.0, CEDS and PRIMAP. Sub sectors 1A and 1B belong to the Energy sector. The total UNFCCC uncertainty is 1.4 % and was*

*calculated based on the UNFCCC NGHGI (2018) submissions. EDGAR v5.0 uncertainties were calculated only for the year 2015 using a lognormal distribution function and is min 3 % and max 4 %.*

$CO_2$ fossil emissions are dominated by the energy sector, which includes emissions from energy use in energy industries (heat and electricity, industry, transport, and buildings). Out of the remaining three sectors (IPPU,

agriculture and waste), IPPU contributes the most to the $CO_2$ emissions, in the EU27+UK these emissions contributed

---

[8] 3G and 3H refer to UNFCCC sector activities, as reported by the standardized common reporting format (CRF) tables, which contain $CO_2$ emissions from agricultural activities: liming and urea applications.





7.1 %, 7.5 %, 5.6 % and 6.4 % from the total NGHGI, EDGAR v5.0 (2017), CEDS (2014) and PRIMAP (2015) respectively. For agriculture and waste, overall, emissions are very small, accounting in the EU27+UK in 2017 for 0.3% (NGHGI) and 0.4 % (EDGAR v5.0) respectively, therefore this difference is negligible for the total C budget.

*Bottom-up estimates by source category*

While Figure 1 was made to assist explanation of differences between datasets disaggregated by sector (e.g., energy industry, transport etc.), in Figure 2 we present $CO_2$ fossil emissions results from EU27+UK split by major source categories (solid, liquid, gas). As in Andrew (2020), we observe good agreement between all data sources and UNFCCC NGHGI (2019) data at this level of regional aggregation. The figure presents estimates for the year 2014,

as that was the most recent year when all sources reported estimates. BP[9] (2019), CEDS (v_2019_12_23), and EDGAR[10] v5.0 (2020) do not publish emissions split by fuel type at the country level and the latter two are shown as dark grey while the former is shown separating gas from liquid/solid.

While the datasets agree well, there are some differences. The EIA (2020) estimate is higher than others, largely because it includes international bunker fuels in liquid-fuel emissions. The IEA (2019) excludes a number of

sources from non-energy use of fuels as well as all carbonates. GCP's total matches the NGHGIs exactly by design but remaps some of the fossil fuels used in non-energy processes from 'Others' to the fuel types used. BP, CEDS, and EDGAR v5.0 all report total emissions very similar to the UNFCCC NGHGI (2019).

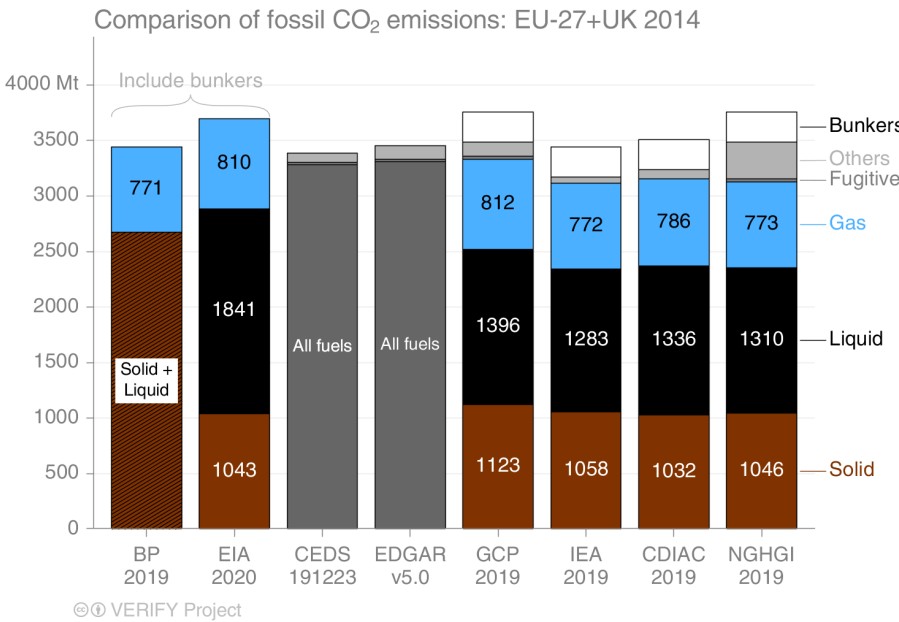

---

[9]For BP, the method description allows for emissions from natural gas to be calculated from BP's energy data, but the data for solid and liquid fuels are insufficiently disaggregated to allow replication of BP's emissions calculation method for those fuels.

[10]EDGAR v5.0 provides significant sectoral disaggregation of emissions, but not by fuel type due to license restrictions with the underlying energy data from the IEA.





*Figure 2: EU27+UK total $CO_2$ fossil emissions, as reported by eight data sources: BP, EIA, CEDS, EDGAR v5.0,*
*GCP, IEA, CDIAC and UNFCCC NGHGI (2019). This figure presents the split per fuel type for year 2014. 'Others'*
*is other emissions in the UNFCCC's IPPU, and international bunker fuels are not usually included in total emissions*
*at sub-global level. Neither EDGAR (v5.0 FT2017) nor CEDS publish a break-down by fuel type, so only the total is*
*shown.*

***Top-down estimates***

        Figure 3 represents the first attempt to evaluate our single inversion of $CO_2$ fossil emissions, based on satellite
CO and $NO_2$ measurements, against BU estimates. The particular inversion reported here provides emission totals for
EU11[11] + Switzerland and these exclude non-fossil fuel emissions (Konovalov et al. 2016, Konovalov & Lvova, 2018).
This inversion estimate partly relies on information available from the BU emission inventories (EDGAR v4.3.2 for
2012                       ((http://edgar.jrc.ec.europa.eu/overview.php?v=432_GHG,
http://edgar.jrc.ec.europa.eu/overview.php?v=432_AP)    and    CDIAC    for    2012-2014    (http://cdiac.ess-
dive.lbl.gov/trends/emis/overview_2014.html) and is therefore not fully independent from BU $CO_2$ fossil emission
estimates. The estimate from the inversion, despite its uncertainty (2700 Tg $CO_2$ (± 480 Tg $CO_2$)) is comparable with
the mean of the $CO_2$ emissions from the NGHGI in 2014 (2624 Tg $CO_2$) and to mean of the other seven BU sources
2588 (±463 Tg $CO_2$). The TD estimate does not include $CO_2$ emissions from cement production while some bottom-
up inventories include them. Cement emissions are known to constitute only a minor fraction (~5 %) of the total fossil
$CO_2$ emissions in Europe (UNFCCC 2019, Andrew et al., 2019, Friedlingstein et al., 2020 in review) and can be
disregarded in the given comparison.

---

[11]The EU11 members are Portugal, Spain, France, Belgium, Luxembourg, Netherlands, United Kingdom, Germany, Denmark, Italy, and Austria

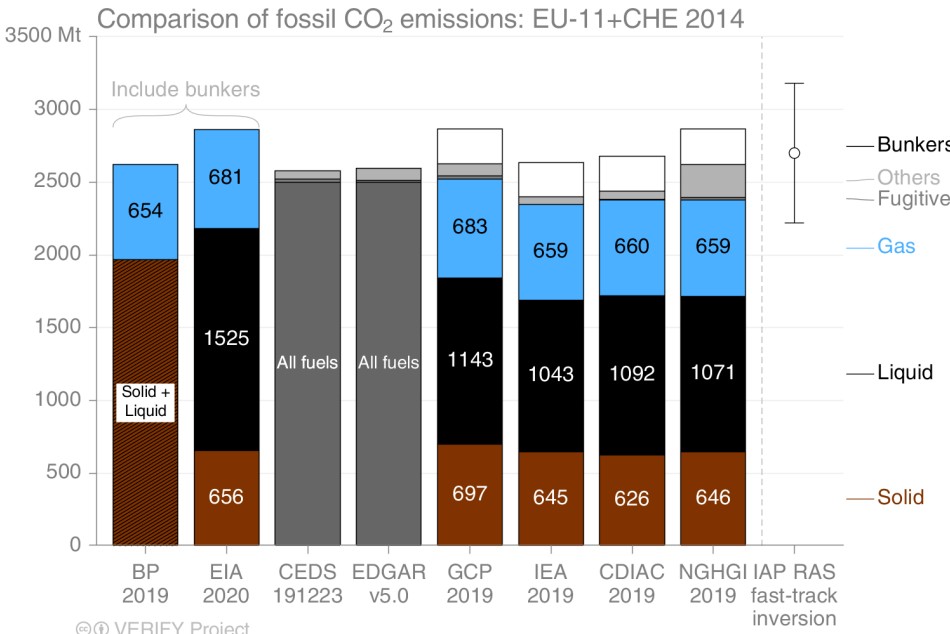

*Figure 3: A first attempt in comparing BU CO$_2$ fossil estimates from eight data sets with a TD fast-track inversion (Konovalov and Lvova, 2018). The data represents EU11 + Switzerland for the year 2014. The uncertainty bar on the inversions represents the 2σ confidence interval.*

### 3.3. CO$_2$ land fluxes

This section presents an update to the benchmark data collection by Petrescu et al., 2020 on CO$_2$ emissions and removals from the LULUCF sector (excluding energy-related emissions, but including emissions from land use change, emissions from disturbances on managed land, and the natural sink on managed land), expanding the scope of that work by adding TD estimates from inverse model ensembles and additional BU models run with higher-resolution meteorological forcing data over the EU27+UK.

Land CO$_2$ fluxes result from CO$_2$ emissions/removals from one land type converted to another (e.g., forests cleared for croplands), as well as emissions/removals from land occupied by terrestrial ecosystems (depending on the dataset, this may be from managed or unmanaged land, which complicates comparisons with NGHGIs). Such fluxes typically include emissions and sinks in soils and carbon shifts due to harvests, including emissions from the decay of harvested wood products (HWP). Some estimates are specific to a given vegetation/sector type (i.e., only cropland or

grassland). As discussed by Petrescu et al., 2020, the analyzed fluxes therefore relate to emissions and removals from direct LULUCF activities (clearing of vegetation for agricultural purposes, regrowth after agricultural abandonment, wood harvesting and recovery after harvest and management) but also indirect LULUCF for CO$_2$ fluxes due to





processes such as responses to environmental drivers (i.e., climate change and $CO_2$ fertilization) on managed land[12].
Additional $CO_2$ fluxes may occur on unmanaged land, but these fluxes are very small. According NIRs, all land in the
EU27+UK is considered managed, except for 5% of France territory.

The indirect $CO_2$ fluxes on managed and unmanaged land, are part of the land sink in the definition used in
IPCC Assessment Reports or the Global Carbon Project's annual global carbon budget (Friedlingstein et al., 2019),
while the direct LULUCF fluxes are termed "net land-use change flux". Grassi et al. (2018) have shown that the
inclusion or exclusion of the indirect sink on managed land in LULUCF is a key reason for discrepancy between
reporting and scientific definitions.

Several studies have already analyzed the European land carbon budget from different perspectives and over
several time periods using GHG budgets from fluxes, inventories and inversions (Lyussaert et al,, 2012), flux towers
(Valentini et al., 2000), forest inventories (Liski et al., 2000, Pilli et al., 2017, Nabuurs et al., 2018) and IPCC
Guidelines (Federici et al., 2015, Tubiello et al., 2020), in addition to the first benchmark data collection of BU
estimates (Petrescu et al., 2020).

Achieving the well-below 2°C temperature goal of the PA requires, among other things, low-carbon energy
technologies, forest-based mitigation approaches, and engineered carbon dioxide removal (Grassi et al., 2018, Nabuurs
et al. 2017). Currently, the EU27 + UK reports a sink for LULUCF and forest management will continue to be the
main driver affecting the productivity of European forests for the next decades (Koehl et al., 2010). For the EU to
meet its ambitious climate targets, it is necessary to maintain and even strengthen the LULUCF sink (COM(2020)
562). Forest management, however, can enhance (Schlamadinger et al., 1996) or weaken (Searchinger et al., 2018)
this sink. Furthermore, forest management not only influences the sink strength, it also changes forest composition
and structure, which affects the exchange of energy with the atmosphere (Naudts et al., 2016), and therefore the
potential of mitigating climate change (Luyssaert et al., 2018; Grassi et al., 2019). Meteorological extremes (made
more likely through climate change) can also affect the efficiency of the sink (Thompson et al., 2020). Therefore,
understanding the evolution of the $CO_2$ land fluxes is critical to meet the goals set out in the Paris Agreement.

### 3.3.1.    Estimates of European and regional total $CO_2$ land fluxes

We present results of total $CO_2$ land fluxes from EU27 + UK and five main regions in Europe: North, West,
Central, East (non-EU) and South. The countries included in these regions are listed in Appendix A, table A.

Figure 4 shows the total $CO_2$ fluxes from NGHGI for both 1990 base year and mean of 2011-2015 period.
We aim with this period to bring together all information over a five year period for which values are known in 2018.
In fact this can be seen as a reference for what we can achieve in 2023, the year of the first global stocktake, where
for most UN Parties the reported inventories will be compiled only up to the year 2021. Given that the GST is only
repeated every 5 years, a five-year average is clearly of interest.

The $CO_2$ fluxes in Figure 4 include direct and indirect LULUCF on managed land. The total UNFCCC
estimates include the total LULUCF emissions and sinks (by the UNFCCC definition) belonging to all six IPCC land
classes and HWP (see section 2.3, Appendix B1, Figure B1b). We plot these and compare them with fluxes simulated

---

[12] In NGHGI reporting, land in EU is considered to be managed.



with statistical global datasets, bookkeeping and biosphere models, sector-specific models and inversion model ensembles. The error bar represents the variability in models estimates as the min and max values in the ensemble.

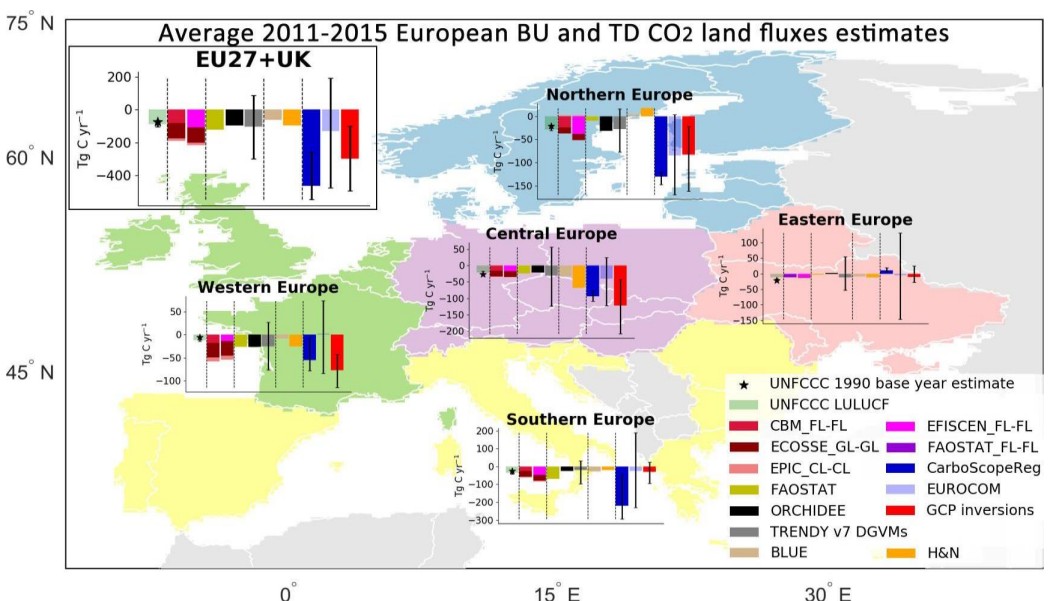


*Figure 4: Five-year average (2011-2015) $CO_2$ land flux estimates (in Tg C) for EU27 + UK and five European regions (Northern, Western, Central, Southern and Eastern non-EU). Eastern Europe does not include European Russia and the UNFCCC uncertainty for the Republic of Moldova was not available. Northern Europe includes Norway. Central Europe includes Switzerland. The data are UNFCCC NGHGI (2019) submissions (grey) and base year 1990 (black*

*star), four sector-specific BU models for FL-FL (CBM, EFISCEN), CL-CL (EPIC-IIASA), and GL-GL (ECOSSE), ecosystem models (ORCHIDEE and TRENDY v7 DGVMs), FAOSTAT, two bookkeeping models (BLUE and H&N), TD inversion ensembles (GCP2018, EUROCOM) and one regional European inversion represented by CarboScopeReg.*

For all regions and the EU27+UK, we note considerable disagreement between the BU and TD results. We mostly see that BU (observation-based and process-based models) agree well with the NGHGIs, while inversions, in particular EUROCOM, report very strong sinks and high variability of the results compared to the BU estimates. We believe that, in general, the differences we see between regions' TD and BU results are linked to model-specific set-ups and definition issues explained in detail in sections 3.3.2 (process-based models and NGHGIs), 3.3.3 (DGVMs,

bookkeeping models and NGHGIs) and 3.3.4 (all BU, TD and NGHGIs). As the current analysis is a first attempt to quantify EU27+UK estimates as a whole, we aim in the future to deepen the analysis for regional/country results.

### 3.3.2. LULUCF $CO_2$ fluxes from NGHGI and decadal changes

In Figure 5 we show the $CO_2$ LULUCF flux decadal change from UNFCCC NGHGI (2019). The contribution

of each category ("remaining" and "converted") to the overall reduction of $CO_2$ emissions in percentages between the

three mean periods (grey columns are the mean values over 1990-1999, 2000-2009 and 2010-2017). The "+" and the "-" signs represent a source and a sink to the atmosphere. LUC(-) are the land use conversion changes that increase the strength of the LULUCF sink between two averages; LUC(+) are the land use conversion changes that decrease the strength of the overall LULUCF sink. Note that the sectors inside LUC(-) may be sources or may be sinks, but

between the two average periods, they become more negative. For the period between 1990-1999 mean and 2000-2009 mean the overall reduction is -9.5 % (i.e., increased land sink) with positive contribution from FL-FL and LUC(+) (wetlands, settlements and other land conversions) contributing to weakening the overall sink (+3.5 %)[13], and with all others conversions contributing to the strengthening of the sink (-13 %)[14]. For the period between the 2000-2009 mean and the 2010-2017 mean we notice that the main contributors to the overall +3.5 % increase are FL-FL,

HWP and LUC (+) (forest, wetlands and settlement conversions) which contribute (+7.2 %) to weakening the sink, while GL-GL, CL-CL and LUC (-) (cropland, grassland and other conversions) contribute to strengthening the sink (-3.7 %).

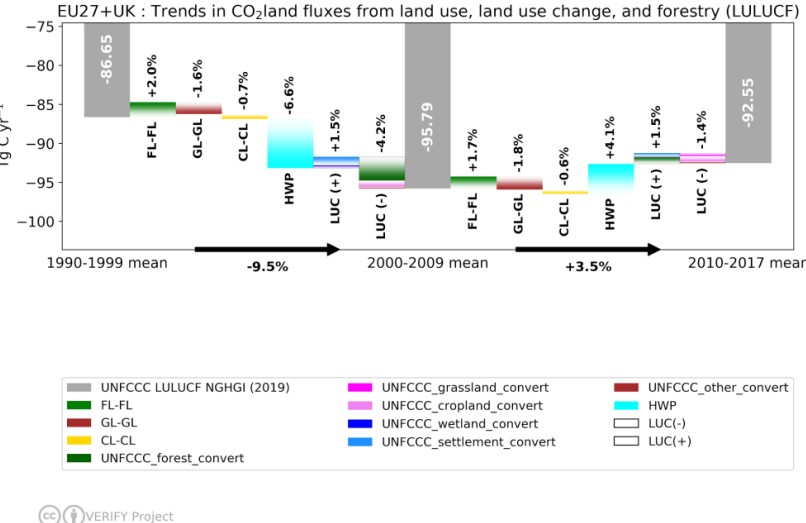

Figure 5: The contribution of changes (%) in various LULUCF categories to the overall change in LULUCF multi-

year mean emissions as reported by Member States to the NGHGI UNFCCC (2019). Changes in land categories converted to other land are grouped to show net gains and net losses in the same column, with the bar color dictating which category each emission belongs to; note that the composition of the "LUC(+)" and "LUC(-)" bars can change between time periods. Not shown are emissions from "Wetlands remaining wetlands", "Settlements remaining settlements", and "Other land remaining other land" as none of the BU models used distinguish these categories. The

---

[13] positive %s - source

[14] negative %s – sink



*fluxes follow the atmospheric convention, where negative values represent a sink while positive values represent a*
*source.*

We see that HWP emissions are by far the major contributor but in different directions across the two periods,
from strengthening the sink between 1990-1999 and 2000-2009 to reducing the sink in the second period. This is
mostly due to the specific accounting approach where a reduction on the amount of harvest, such as the one occurred
after the economic crisis in 2008, progressively reduced the inflow of raw material and, taking into account of the
decay rate applied to each commodity, this further reduced the C stock within the same pool. Therefore, Figure 5
suggests that carbon emission from HWP decay became greater than the amount of carbon entering HWP in recent
decades.

### 3.3.2. Estimates of CO$_2$ fluxes from bottom-up approaches

In this section we present annual total net CO$_2$ land emissions between 1990-2018 i.e., induced by both
LULUCF and other (environmental changes) processes from class specific models as well as from models that
simulate some or all classes. The definitions of the classes might differ from the definition of the LULUCF (FL, CL,
GL etc.) (Figures 6, 7 and 8) where, according to IPCC 2006 guidelines, to become accountable in the NGHGI under
remaining categories, a land-use type must be in that class for at least 20 years. Over FL (both FL-FL and conversions)
we compare modelled net biome production (NBP) estimates (including soil plus living and dead biomass C stock
change) simulated with class-specific ecosystem models to UNFCCC and FAOSTAT data consisting of net carbon
stock change in the living biomass pool (aboveground and belowground biomass) associated with forests and net
forest conversion including deforestation.

The **Forest Land** estimates, which remain in this class (FL-FL) in Figure 6, were simulated with ecosystem
models (CBM, ORCHIDEE, EFISCEN) (described in Appendix A2 and Appendix Table B1), global datasets
(FAOSTAT) and countries' official inventory statistics reported to UNFCCC. The results show that the differences
between models are systematic, with CBM having slightly weaker sinks than EFISCEN and FAOSTAT. Starting with
year 2000 and towards 2017, the FAOSTAT reports sinks that strengthen over time. Differences between estimates
might be due to the use of different input data e.g. CBM and EFISCEN use national forest inventory (NFI) data as the
main source of input to describe the current structure and composition of European forest, while FAOSTAT uses input
data directly from country submission done under FAO Global Forest Resource Assessments (FRA[15]) (e.g. carbon
stock change calculated by FAO directly from carbon stocks and area data submitted by countries directly).
Furthermore, FAOSTAT numbers include afforestation, i.e., the sum of all other land converted to FL, resulting in a
smaller sink if afforestation would be removed, therefore matching better the UNFCCC estimates (Petrescu et al.,
455 2020).

For ORCHIDEE, the model shows a high inter-annual variability in carbon fluxes because ORCHIDEE
operates on a sub-daily time step for most biogeochemical and biophysical processes except a daily time step for
"slow" processes like carbon allocation in the vegetation reservoirs, while all other models involved in this comparison

---

[15]The Global Forest Resource Assessment (*FRA*) is the supplementary source of Forest land data disseminated in *FAOSTAT,*
http://www.FAO.org/forestry/fra/en/



use forest inventory data which is reported every few years (i.e., five years for FRA). ORCHIDEE results indicate that

climatic perturbations and extreme events (multi-month droughts, in particular) can have significant impacts on the net carbon fluxes depending on when they occur. This is to some extend supported by dendrometer data although highly varying per site and tree species obscuring a significant net effect (Scharnweber et al., 2020). It should also be noted that dendrometer data measures carbon stored in individual trees, while the NBP reported in figures in this paper include fluxes from litter and soil respiration. The variability of the weather data affects all components of the carbon

dynamics in the ecosystems (hence NBP), with for instance impacts on C assimilation rates, length of the growing season, dynamics of respiration rates and allocation of the carbon in the plant (cf. Figure 1 and 2 in Reichstein et al. (2013)).

The UNFCCC NGHGI uncertainty of $CO_2$ estimates for FL-FL across the EU27+UK, computed with the error propagation method (95% confidence interval) (IPCC, 2006), ranges between 23 % - 30 % when analyzed at the

country level as it varies as a function of the component fluxes (EU NIR, 2017). Given the different methodologies and input data for emission calculation and uncertainties in each method (10 Tg C yr$^{-1}$ for the mean), we consider the match between the model EFISCEN and the UNFCCC NGHGI (2019) estimates to be good, in particular with respect to the similarity in temporal trends. The means of ORCHIDEE and CBM fall within the reported UNFCCC uncertainty (around 20 Tg C yr$^{-1}$), while FAOSTAT lies outside of it. Note that FAOSTAT and EFISCEN have a different trend

compared to other models and the NGHGI.

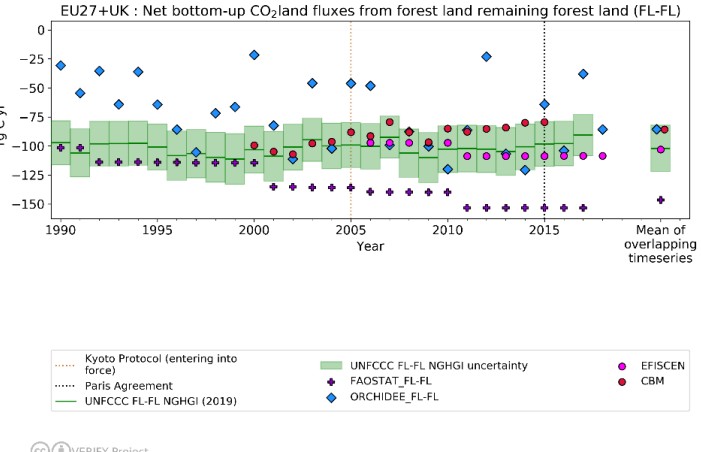

*Figure 6: Net $CO_2$ land flux from forest land remaining forest land (FL-FL) estimates for EU27+UK $CO_2$ from UNFCCC NGHGI 2019 submissions and bottom-up emission models with their 1990-2017 mean (on the right side). CBM FL-FL estimates include 25 EU and UK countries (excl. Cyprus and Malta); The relative error on the UNFCCC*

*value represents the UNFCCC NGHGI (2018) MS-reported uncertainty computed with the error propagation method (95% confidence interval) and is 19.6 % (with no values for Hungary and Cyprus). The negative values represent a sink.*





Some of the reasons for differences between estimates we see in Figure 6 are linked to different activity data (e.g. forest area) the models use, for example the stronger sink reported by FAOSTAT compared to UNFCCC NGHGI. By analyzing three of the forest area products (ESA-CCI LUH2v2 (Hurtt et al., 2011) used in ORCHIDEE, FAOSTAT and UNFCCC) we found the following:

- ESACCI LUH2v2, based on a combination of the ESA-CCI land cover map for 2015 with the historical land cover reconstruction from LUH2 (Lurton et al., 2020), assumes that the shrub land cover classes are equivalent to forest. In terms of area, globally 2.5% of the land in the CCI product is shrub land, and taking into account that forest in Europe represents one third of the land, a change of ~2% would be equivalent to a 10% difference.
- ESACCI LUH2v2 does not include the 20-year transition period, as included in the IPCC reporting guidelines. This could be 1% of the forests in Europe, but there is a considerable uncertainty in that based on the transition data seen between the maps.
- FAOSTAT Forest Land area does not include only forest remaining forest but only the time series of FL, forest land. That means it includes afforestation but not deforestation. This area is based on the same land use/land cover maps, hence relying on net land use changes.

**Cropland and Grassland** (CL and GL) (in UNFCCC NGHGI 2019, UNFCCC sectors 4B and 4C, respectively) include net $CO_2$ emissions/removals from soil organic carbon (SOC) under 'remaining' and 'conversion' categories. Similar to Forest Land, we present in Figure 7 the fluxes belonging to the 'remaining' category CL-CL. The cropland definition in the IPCC includes cropping systems and agro-forestry systems where vegetation falls below the threshold used for the forest land category, consistent with the selection of national definitions (IPCC glossary).

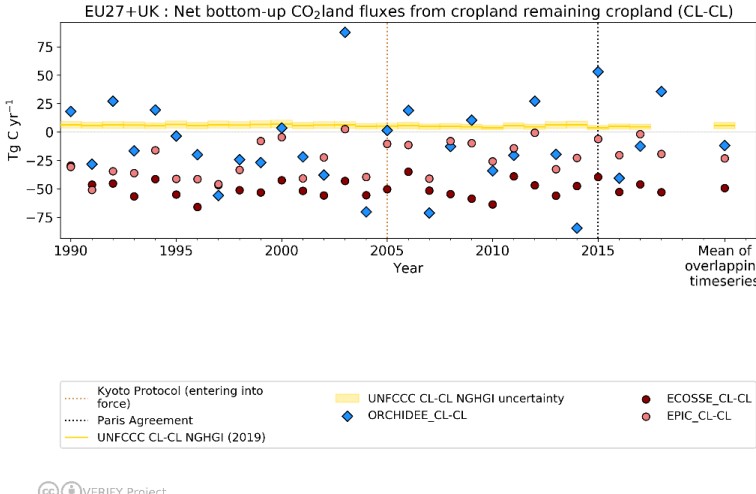

*Figure 7: Net $CO_2$ flux from cropland remaining cropland estimates for EU27+UK from UNFCCC NGHGI 2019 submissions and bottom-up emission models with their 1990-2017 mean (on the right side). CL-CL emissions*





From Figure 7 we see that modelled CL-CL inter-annual variabilities simulated by ECOSSE and EPIC-IIASA estimates are consistent, while ORCHIDEE shows a much larger year-to-year variation. The NGHGIs are

mostly insensitive to inter-annual variability as the estimations are mainly based on statistical data for surfaces/activities and EFs that do not vary with changing environmental conditions.

The three process-based models report sinks in most years (means of -12, -49 and -23 Tg C respectively), contrary to the NGHGIs, which report a small but constant source over the whole period (mean of $5.6 \pm 3.5$ Tg C) with almost no inter-annual variability by construction. The source reported by NGHGIs, at EU level, is mostly

attributed to emissions from cropland on organic soils[16] in the northern part of Europe which emit $CO_2$ due to C oxidation from tillage activities. As an example, Finland and Sweden report together more than half of the total area of organic soil in Europe. Organic soils are an important source of emissions when they are under management practices that disturb the organic matter stored in the soil. In general, emissions from these soils are reported using country-specific values when they represent an important source within the total budget of GHG emissions. In the

southern part of Europe, the two categories (CL-CL and GL-GL) are a sink, due to a lack of organic soils in those regions and due to an abandonment trend of land converting arable land to grassland (EU NIR, 2019). In addition, NGHGIs assume that all aboveground biomass of non-woody crops re-enters the atmosphere at harvest. In models like ORCHIDEE and EPIC-IIASA, only part of the aboveground biomass is harvested and enters the atmosphere, and the rest (approximately 50% of the aboveground carbon) enters the soil and decays. Given more favorable growing

conditions due to climatic changes and $CO_2$ fertilization, this can lead to more carbon entering the soil in ORCHIDEE in recent decades, which is driving the CL-CL sink observed in the model.

The strongest sink reported by ECOSSE model is linked to the soil C model (RothC) used, which simulates a large 'inert pool' which thus leads to a slower C turnover time in the soil (compared to ORCHIDEE or EPIC-IIASA) and thus to significantly larger sink. This 'respiration' aspect of RothC will be addressed in the next synthesis.

According to Ciais et al., 2010, a small carbon source would be a realistic assumption for croplands and in line with the NGHGI report. Thus, while the NGHGI and the three process-based models show a different sign of the $CO_2$ flux, the difference is a result of the processes included/definitions used in each approach, as explained above.

For the inter-annual variability all three models follow the same dynamic, but the impacts of climate extremes are different with significantly larger impacts in ORCHIDEE. While ORCHIDEE shows a strong reaction to drought

impacts changing from a sink to a source (e.g. for 2003, which is reported as a very dry year (Ciais et al., 2005)), the other two models follow ORCHIDEE's variation trend, but show less extremes. As ECOSSE directly simulates the

---

[16]The 2006 IPCC Guidelines largely follow the definition of Histosols by the Food and Agriculture Organization (FAO), but have omitted the thickness criterion from the FAO definition to allow for often historically determined, country-specific definitions of organic soils (see Annex 3A.5, Chapter 3, Volume 4 of the 2006 IPCC Guidelines for National Greenhouse Gas Inventories (2006 IPCC Guidelines) and Chapter 1, Section 1.2 (Note 3) of 2013 Supplement to the 2006 IPCC Guidelines for National Greenhouse Gas Inventories: Wetlands (Wetlands Supplement, IPCC 2014): https://www.ipcc-nggip.iges.or.jp/public/wetlands/pdf/Wetlands_separate_files/WS_Chp1_Introduction.pdf)).

Earth System
Science
Data

annual net primary production (NPP) (i.e., internal component model (MIAMI) implemented in ECOSSE) and not the intra-annual gross primary production (as in ORCHIDEE), the impact of season specific climate anomalies is smaller than in ORCHIDEE.

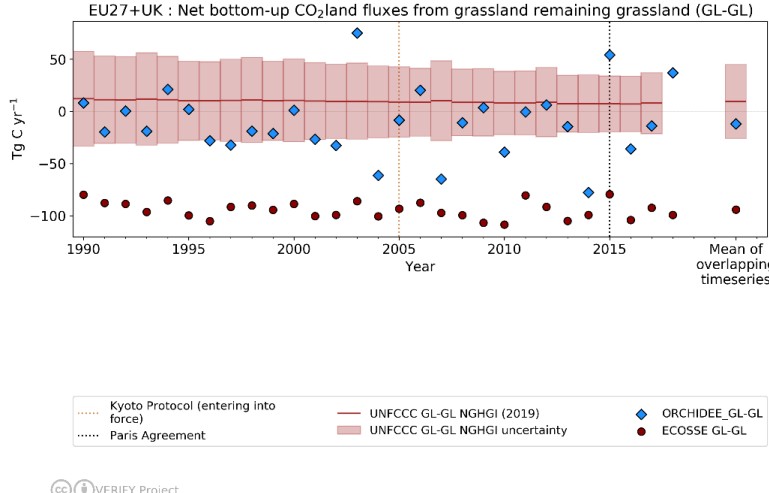


Figure 8: Net $CO_2$ flux estimates from grassland remaining grassland for EU27+UK $CO_2$ from UNFCCC NGHGI 2019 submissions and bottom-up emission models with their 1990-2017 mean (on the right side). GL-GL emissions are estimated with the ORCHIDEE and ECOSSE models. The relative error on the UNFCCC value represents the UNFCCC NGHGI (2018) MS-reported uncertainty computed with the error propagation method (95% confidence interval), and is equal to 373.6 % (no data for Hungary, Cyprus, Slovakia, Spain and Czech Republic. The negative values represent a sink, while the positive represent a source.

Figure 8 shows the $CO_2$ flux of the grassland remaining grassland category, GL-GL. Grassland definition in the IPCC includes rangelands and pasture land that is not considered as cropland, as well as systems with vegetation that fall below the threshold used in the forest land category. This category also includes all grassland from wild lands to recreational areas as well as agricultural and silvo-pastural systems, subdivided into managed and unmanaged, consistent with national definitions (Petrescu et al., 2020). The NGHGIs of countries in the EU-27+UK report emissions from managed pastures only, which, in 2010, represented a minimum of 58 % (Chang et al., 2016) of the total managed grassland area in the EU. Since almost all European grasslands are somehow modified by human activity and have to a major extent been created and maintained by agricultural activities, they could be defined as "semi-natural grasslands", even if their plant communities are natural (EU LIFE, 2008). Therefore, NGHGIs report a small mean source over 1990-2017 (9 Tg C) primarily due to the use of EFs from national statistics which are linked to intensive management practices applied to grasslands in the EU.

Out of all the models used in this study, only ORCHIDEE and ECOSSE report fluxes from this category. Grasslands in ORCHIDEE do not undergo any specific management and are not separated from pasturelands.





Therefore, discrepancies between ORCHIDEE and the NGHGI data result in the first reporting a mean sink over 1990-2017 of -12 Tg C while official inventories report a small source, as explained above. The sink in ORCHIDEE is due the fact that the $CO_2$ fertilization effect increases the NPP over time and also increases input of C to the soil, which then leads to increased soil C stocks. The strong sink simulated by ECOSSE (-94 Tg C in mean) is the result of using

a limiting scenario where intensively managed grasslands, i.e., high grazing intensity and high yield removal, are not included, thus favoring high soil carbon storage. These effects are similar to that seen in croplands (see above), resulting from the $CO_2$ fertilization effect.

### 3.3.3. Bottom-up CO₂ estimates from all LULUCF sectors


In this section we attempt to present a comprehensive analysis of $CO_2$ emissions and sinks for the LULUCF sectors. Here we try to compare the sum of all categories and sectors of the NGHGIs discussed in Figure 5 (including the remaining and transition sub-sectors; details are found in the Figure 5 caption), with various observation-based BU model estimates. The comparison with atmospheric inversions (TD) is discussed in the next section. Such

comparison is challenging due to differences in terms of activities covered in the different estimates, as well as differences in terminology, which have already been highlighted in several papers (see more specifically Petrescu et al., 2020, Figure 12). Let's first briefly recall the main differences between the selected products:

- FAOSTAT differs from NGHGIs for reasons summarized by Federici et al 2015 and Petrescu et al 2020, including numerically different data provided by Member States to FAOSTAT and UNFCCC, different

585        methods (FAOSTAT applies a Tier 1 approach globally, while Member States reports to the UNFCCC vary from Tier 1 to Tier 3), differences between net and gross land use (FAOSTAT is based on net transitions), and FAOSTAT results only considering living biomass pools instead of the five IPCC pools[17] reported to the UNFCCC.

- The process-based high resolution ORCHIDEE simulation and the TRENDY v7 ensemble, with the so-called

590        "S3 simulation" (see the TRENDY simulation protocol, Le Quéré et al. 2018), include the impact of $CO_2$ fertilization, climate change and land use change for forest, grassland and cropland sectors; they do not explicitly treat the wetland, settlement and other land sectors as in the NGHGIs. They account for the evolution of living and dead biomass as well as SOC for all categories while for NGHGIs it is not mandatory for all subcategories (i.e., dead biomass). Finally, there is significant uncertainty associated to the DGVMs'

595        fluxes both from i) the forcing data, including datasets of land-use changes and the coverage of different land use change practices, ii) model parameters, and iii) structural uncertainty in models (i.e., which processes are included and which are not) (Arneth et al 2017). Similar to FAOSTAT, DGVMs typically deal with net land use change emissions, instead of gross land use change as reported in NGHGIs, which may induce significant differences with coarse resolution model simulations (i.e., 0.5° or 1° for the TRENDY ensemble). DGVMs

---

[17] According IPCC 2006 guidelines the reporting is done for the five LULUCF carbon pools: above-ground biomass, belowground biomass, dead wood, litter, and soil organic matter

often do not distinguish between managed and unmanaged land, while NGHGIs are for emissions from
        managed land.

- The bookkeeping models, BLUE and H&N, calculate net emissions from land use change including immediate emissions during land conversion, legacy emissions from slash and soil carbon after land-use change, regrowth of secondary forest after abandonment, and emissions from harvested wood products when
they decay. They thus do not account for the net fluxes occurring in the "remaining" land categories due for instance by the $CO_2$ fertilization effects or climate changes. One exception to this are fluxes from wood harvested, which is a primary source of emission on managed forest land and also included in bookkeeping models. As seen before in Figure 5, this component can present a significant flux.

Given all these differences in terms of activities, the comparison in this section should be considered as a first step that raises both important aspects of the C cycle and questions that need to be addressed in the future. Going toward a more specific comparison of only net land-use change fluxes would require additional considerations. In GCP's annual global carbon budget, this term is estimated by global DGVMs as the difference between a run with and a run without land-use change and by bookkeeping models. Such an estimate is given in Figure 13 in Petrescu et
al., 2020 for forest land. While attractive, such a plot does not fully resolve the differences mentioned above. In particular, questions remain about net vs. gross land use change, managed vs. unmanaged land, and emissions from wood harvest. In addition, UNFCCC "convert" emissions (i.e., emissions resulting from land that has been converted from one type to another) are calculated for 20 years following conversion. FAOSTAT, DGVMs, and bookkeeping models typically only include "convert" fluxes from the year following conversion, although bookkeeping models can
more easily include this transition period.

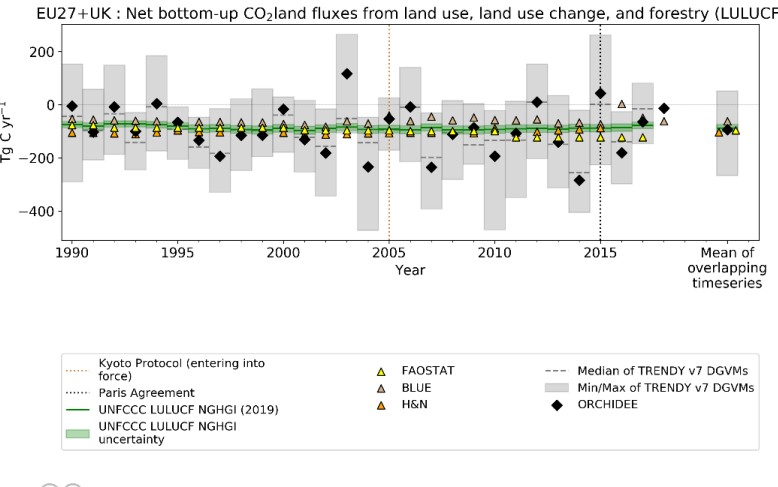

*Figure 9: A comparison of different estimates of the CO₂ fluxes from land use, land use change and forestry activities in the EU27+UK from seven data sources: UNFCCC NGHGI (2019), BLUE, H&N, DGVMs (TRENDY v7), FAOSTAT*





*and ORCHIDEE (stand-alone with high spatial resolution forcing and from TRENDY). The grey bars represent the*
*individual model data for eight DGVMs. The UNFCCC estimate includes the following categories: Forest Land,*
*Cropland, Grassland, Wetlands, Settlements, and Other Land from conversions, in addition to harvested wood*
*products (HWP). The relative error on the UNFCCC value represents the UNFCCC NGHGI (2018) MS-reported*
*uncertainty computed with the error propagation method (95% confidence interval) and is 16 %. The FAOSTAT*
*estimate includes the following categories: Forest Land, incl. afforestation and deforestation as conversion of forest*
*land to other land types). The means are calculated for the 1990-2015 overlapping period. The negative values*
*represent a sink, while the positive represent a source.*

Figure 9 thus represents $CO_2$ fluxes from LULUCF activities, including estimates from ORCHIDEE high resolution and TRENDY (mean across the ensemble) DGVMs models ("S3" type simulations), bookkeeping models,
NGHGI and FAOSTAT. For the overlapping period 1990-2015, we observe from the means (see right part of the plot) that both bookkeeping models (BLUE (-61 Tg C) and H&N (-103 Tg C)) and FAOSTAT (-96 Tg C) estimates match the UNFCCC NGHGI (-87 Tg C) reporting, because their managed areas for EU27+UK are similar (H&N: 118 Mha; BLUE: 117 Mha; UNFCCC: 167 Mha, from in Grassi et al., 2018a, Petrescu et al., 2020). Unmanaged area in the EU27+UK is negligible and sums up only 4 Mha. The similarities between bookkeeping models and UNFCCC can
be explained by the fact that, despite a smaller forest sink in H&N, they both report a small sink in non-forest land uses while for these land uses UNFCCC reports a source (Figures 7 and 8).

The UNFCCC LULUCF estimates contain $CO_2$ emissions from all six land use classes and HWP, including remaining classes and conversion to and from a class to another. ORCHIDEE (-93.9 Tg C) shows large variabilities (black diamonds), mostly following the temporal patterns of the mean from TRENDY v7 DGVMs (-103 Tg C) (grey
bars) as detailed above. Note again that ORCHIDEE is also part of the TRENDY ensemble, but with a different meteorological forcing (coarser resolution, 0.5°) than the one used within the VERIFY project (around 0. 1° resolution).

The differences between bookkeeping models and UNFCCC and FAOSTAT are discussed in detail in Petrescu et al., 2020 cf. Fig. 12, who concludes that the key difference between bookkeeping models, on the one hand,
and FAOSTAT and UNFCCC methodologies, on the other, is that the latter are based on the managed land proxy (Grassi et al., 2018a). ORCHIDEE model and the TRENDY v7 ensemble means show much higher inter-annual variability due to the sensitivity of the model fluxes to highly variable meteorological forcing and the models' sub-daily time steps which allow for much more rapid responses to changing conditions (i.e. 2003 extreme drought year), as already discussed in the previous sections. The incorporation of variable climate data and the fact that DGVM
models simulate explicitly climate impacts on $CO_2$ fluxes, which inventories and bookkeeping do not, explain these differences.

DGVMs estimate net land-use emission as the difference between a run with and a run without land-use change, and their estimate includes the loss/gain of additional sink capacity, that is, the sink that favors the environmental changes (e.g. $CO_2$ fertilization). This sink created over forest land in the simulation without land use
change is "lost" in the simulation with land use change (i.e., deforestation) because agricultural land lacks the woody



material and thus has a higher carbon turnover (Gasser et al., 2013, Pongratz et al., 2014 and cf. Figure 12 in Petrescu et al., 2020). This different definition from bookkeeping models historically implies on average higher carbon 'land use' emissions from DGVMs when an ecosystem is converted to another with a lower carbon density, even if all post-conversion carbon stocks changes were the same in DGVMs and bookkeeping models.


### 3.3.4. Comparison of top-down and bottom-up $CO_2$ estimates

Figure 10 highlights the variability of estimates from atmospheric inversions of GCP (1990-2017), CarboScopeRegional (2006-2017) and EUROCOM (2006-2015) plotted against total annual EU27+UK $CO_2$ land emissions/removals from observation-based BU approaches and UNFCCC NGHGI (2019). In these inversions, all components of the carbon cycle (NEE) that contribute to the observed atmospheric $CO_2$ gradients between stations (e.g., lateral fluxes, oxidation of C compounds into $CO_2$) are included. To facilitate the comparisons with NGHGIs we first account for some of these differences by subtracting from the inversion estimates of the emissions from rivers (Lauerwald et al., 2015), lakes and reservoirs (Raymond et al., 2013; Hastie et al., 2019) as NGHGIs do not include them. Also, not included in NGHGI estimates are the outgassing from crop and wood products traded and consumed this year.

Looking at TD estimates, the annual mean (overlapping period 2006-2015) of the EUROCOM inversions (-138 Tg C) is the closest inversions ensemble (among the three) to the time series mean of the NGHGI estimates (-90 Tg C), with a difference of 48 Tg C yr$^{-1}$ that is well within the mean uncertainty of the regional inversion ensemble (about 250 Tg C yr$^{-1}$). It also matches well with the TRENDY v7 DGVMs trend which is smaller (+7.3 Tg C yr$^{-2}$) than that of the global GCP inversions (-16 Tg C yr$^{-1}$). On the other hand, the large range of variability in the EUROCOM ensemble estimates (+ 335 Tg C in 2015 to – 615 Tg C in 2013) demonstrates that there is still a very significant uncertainty in the TD estimates. This variability seen from the TD estimates is primarily due to uncertainties in atmospheric transport modeling, boundary conditions and uncertainty inherent to the limitation of the observation network.

Additional analyses are still ongoing with the different inversion ensembles to analyze the factors controlling the large difference obtained when compared to BU approaches (for instance, the effect of the a priori fluxes, observation sites, a priori flux and observation uncertainties, and boundary conditions). This paper should be taken cautiously as a first comparison at a spatial scale not investigated so far (i.e., EU27+UK).

The GCP results show a clear trend towards increasing the $CO_2$ sink strength of the land surface in later years, contrary to the NGHGI estimates, which are relatively stable. Thus, the initially reasonable agreement between the two datasets (2000-2005) becomes a difference well outside the uncertainty range of the NGHGI in 2017 (290 Tg C difference between GCP and NGHGI, with an NGHGI uncertainty of only 30 Tg C). Between 2011 and 2018, GCP (-241 Tg C mean) (red bars) shows, as well as large inter-annual variability, an increase in the $CO_2$ sink. The strongest sink between inversions (mean -381 Tg C) is reported by CarboScopeReg which, similar to EUROCOM, also shows high fluctuations. This fluctuation partly reflects the fact that all other inversions are results from ensembles of inversion systems each with different inter-annual variations, while CSR is a single inversion system (just a small ensemble with differing prior error structure and different set of atmospheric station data used).

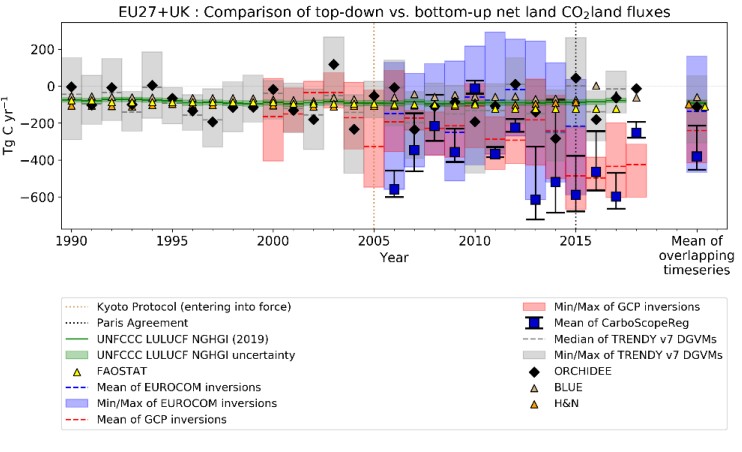

**Figure 10:** *Comparison of BU and TD total EU27+UK biogenic $CO_2$ estimates. The green line represents the UNFCCC NGHGI 2019. The BU estimates belong to bookkeeping models (BLUE, H&N), the grey shade is the DGVMs TRENDY v7 and we plot separately ORCHIDEE and FAOSTAT (FRA) data. The TD estimates belong to models from the ensembles GCB 2019 (red), EUROCOM (blue) and CarboScopeReg (box with whiskers). The relative error on the UNFCCC value represents the UNFCCC NGHGI (2018) MS-reported uncertainty computed with the error propagation method (95% confidence interval) and is 16 %. The time series mean overlapping period is 2006-2015. The colored area represents the min/max of model ensemble estimates. The negative values represent a sink, while the positive represent a source. In Appendix B, Figure B1c we show the expanded figure of the mean time series.*

Also, noteworthy is that the global inversions provide reliable results at a global scale (following the atmospheric global $CO_2$ growth rate) but the ranges of estimates when considering continental to regional scales increase significantly due to the difficulties of the inversion systems to separate regional fluxes (e.g. Friedlingstein et al 2020). Note also that these systems are still primarily designed for large scale flux estimates (for instance the CarboScope global system uses a transport model at coarse spatial resolution (4° x 5°) and an error correlation length of 1000 km over land). The regional inversions (EUROCOM and CarboScopeReg) are still systems in development with additional complexity due to the treatment of the boundary fluxes (compared to the global systems).

For the models, differences result from choices in the simulation setup and depend on the type of model used – bookkeeping models, DGVMs, or inventory-based – and whether fluxes are attributed to LULUCF emissions due to the cause or place of occurrence (indirect fluxes on managed land included in NGHGIs and FAOSTAT). Table 3 below highlights these differences by presenting an overview of processes included in the models, seen for the moment as the main cause of discrepancies between estimates shown in Figure 10.



*Table 3: Comparison of the processes included in the inventories, bottom-up models and inversions.*

| Description | NGHGI | | Process-based models | | | | DGVMs | | Bookkeeping Models | | Inversions | | |
|---|---|---|---|---|---|---|---|---|---|---|---|---|---|
| | UNFCCC[a] | FAOSTAT[a] | ECOSSE | EPIC-IIASA | CBM | EFISCEN | TRENDY v7 | ORCHIDEE | BLUE | H&N | CarboScope Regional | Eurocom* | Global Inversions (GCP)** |
| **Ecosystem split / LC transitions** | | | | | | | | | | | | | |
| Forest total | E | E | N | N | E | E | Acc. table A1 in GCB 2019 (Friedlingstein et al 2019) | E | E[h] | E[h] | E | E | Acc. Table A3 in GCB 2019 (Friedlingstein et al 2019) |
| Split FL-FL / FL-X / X-FL | E | E | N | N | E | E/I[c]/I | | E | E[h]/E/E | E[h]/E/E | N | N | |
| Cropland total | E | N | E | E | N | N | | E | E[h] | E[h] | E | E | |
| Split CL-CL / CL-X / X-CL | E | N | E | E/N/N | N | N | | E | N/E/E | N/E/E | N | N | |
| Grassland total | E | N | E | N | N | N | | E | E | E | E | E | |
| Split GL-GL / GL-X / X-GL | E | N | E | N | N | N | | E | N/E/E | N/E/E | N | N | |
| Peatland accounting | E | E | N | N | N | N | | N | N | N | N | N | |
| **"Natural" Processes** | | | | | | | | | | | | | |
| $CO_2$ fertilization | N | | N | E | N | N | Acc. table A1 in GCB 2019 (Friedlingstein et al 2019) | E | N[i] | N[i] | N | E | Acc. Table A3 in GCB 2019 (Friedlingstein et al 2019) |
| Climate induced impacts | N | | N | E[f] | I[b] | I | | E | N[i] | N[i] | E | E | |
| Natural disturbances (fires, insect, wind) | N | | N | N | E | N | | N | N[i] | N[i] | N | N | |
| Soil Organic C dynamic | I | | E | E | E | E | | E | N | N | E | E | |
| Lateral C transport (river) | N | | N | N | N | N | | N | N | N | N | N | |





| Direct human-induced processes | | | | | | | | | | | | |
|---|---|---|---|---|---|---|---|---|---|---|---|---|
| Flux from Harvested Wood Products | E | | N | N | I | N[d] | Acc. table A1 in GCB 2019 (Friedlingstein et al 2019) | E | E | E | N | N | Acc. Table A3 in GCB 2019 (Friedlingstein et al 2019) |
| Flux from Crop/Grass harvest | ? | | E | E[e] | N | N | | E | I[i] | I[i] | N | N | |
| Biomass burning | E | E | E | N[g] | E | N | | N | E[j] | E[j] | N | E | |
| N fertilization (with N dep) | I | N | E | N | N | N | | N | N | N | N | N | |
| Flux from drained organic soils | I | E | E | N | I | N | | I | E[j] | E[j] | N | N | |
| **Resolution** | | | | | | | | | | | | | |
| Spatial | Country totals | Country | 0.125 x 0.125 | 0.125 x 0.125 degree | Country totals | Country totals | Acc. table A1 in GCB 2019 (Friedlingstein et al 2019) | 0.125 x 0.125 degree | 0.25 x 0.25 degree | Country totals | 0.5° x 0.5° | 0.5° x 0.5° | |
| Temporal | Yearly, t-2 | Yearly | Yearly | Monthly | Yearly | Every 5 years | | Monthly | Yearly | Yearly | 3-hourly | from 3-hourly to monthly | |

Not included : **N**, Explicitly modeled : **E**, Implicitly modeled: **I**, Partly modeled : **P**

[a]UNFCCC and FAOSTAT are ensemble of country estimates calculated with specific methodology for each country, following some guidelines.
[b]The climate effects can be estimated indirectly by CBM, using external additional input provided by other models
[c]EFISCEN can add this as a scenario variable, there is no internal module that allocates how much forest area there should be.
[d]EFISCEN has only production in m3 but doesn't have a direct HWP module.
[e]Crop yield and residue harvest from cropland (20% of residues harvested in case of cereals, no residue harvest for other crops).
[f]EPIC-IIASA partly accounts for soil drought, i.e., plant growth limitation due to a lack of water in the soils. Heat stress and floods are not accounted for, though.
[g]In principle, burning of crop residues on cropland can be explicitly simulated by EPIC-IIASA. However, not done for VERIFY as it is not a relevant scenario for the business as usual cropland management in Europe.'
[h]forest/cropland/grassland exist and have carbon stocks, but have carbon fluxes only through change to management. FL-FL includes all land-use induced effects (harvest slash and product decay, regrowth after agric abandonment and harvesting)
[i]implicit by using observation-based carbon densities that reflect harvest/climate/natural disturbances
[j]peat burning and peat drainage are not bookkeeping model output, but are added from various data sources during post processing
[*]According Table 2 in Monteil et al.., 2020 and Table A3 in Friedlingstein et al., 2019

## 4. Summary and concluding remarks

The overview and variety of data products described in this study is the first of a series of European $CO_2$ synthesis papers presenting and investigating differences between UNFCCC NGHGIs, bottom-up data-based





inventories, high resolution observation-based BU models, and TD approaches represented by both global and regional inversions.

The $CO_2$ fossil emissions dominate the anthropogenic $CO_2$ flux in the EU27+UK. Fossil $CO_2$ emissions are more straightforward to estimate than ecosystem fluxes. Different BU methods have only minor differences with respect to the NGHGI. These differences can often be attributed to different definitions or assumptions about activity

data or emission factors or by the allocation of fuel types to different sectors (see Fig. 2, section 3.2). Currently, TD methods, albeit only a single inversion using CO/NOx proxies to determine $CO_2$ fossil emissions, show broad agreement with the BU estimates. The TD inversion is not yet capable of verifying the minor differences between the BU estimates. However, a substantial decrease in the level of uncertainty is expected in the near-term with the large-scale deployment of observation networks dedicated to detecting fossil fuel emissions (e.g., with launch of the $CO_2M$[18]

constellation in 2025, Maenhout al., 2020). In the short-term, methodological improvements and the potential co-assimilation of existing $CO_2$ satellite data are also expected to lead to significant decreases in the uncertainty.

The $CO_2$ land fluxes belong to the LULUCF sector, which is one of the most uncertain sectors in UNFCCC reporting due in part to the fact that these fluxes can be either sinks or sources. The IPCC guidelines prescribe methodologies that are used to estimate the $CO_2$ fluxes in the NGHGI, but differences between countries continue to

exist due to the use of specific national circumstances (as permitted under the 2006 IPCC guidelines). When we analyzed the estimates from multiple BU sources (inventories and models) we observe similar sources of uncertainties: (a) differences due to input data and structural/parametric uncertainty of models (Houghton et al., 2012) and (b) differences in definitions (Pongratz et al., 2014; Grassi et al., 2018b, Petrescu et al., 2020). More accurate estimates for LULUCF data will be needed in the post-2020 reporting for the EU27 and UK since the LULUCF sector will now

contribute to the EU's 2030 targets. To better assess natural variability and trends we believe a reconciliation of BU and TD estimates should focus on clearly defined activities over a given period (e.g. 5 years) and regions as presented in Figure 4. The considerable differences in the agreement between BU and TD estimates from regional split are related to areas and for some regions (e.g., Eastern Europe) sparseness of observation data. Regarding the detailed sector-specific and inversion results (Figures 6,7,8,9 and 10) often differences come from choices in the simulation

setup and depend on the type of model used – bookkeeping models, DGVMs, inventory-based or inversion ensembles. Results also differ based on whether fluxes are attributed to LULUCF emissions due to the cause (e.g., direct or indirect) or place of occurrence. For example, indirect fluxes on managed land are included in NGHGIs and FAOSTAT, while additional sink capacity (e.g. Petrescu et al, 2020) is included in estimates from process-based models (e.g., ORCHIDEE or TRENDY DGVMs). A more in depth analysis of regional/country level is foreseen as

part of the overall long-term VERIFY's objectives.

All observation-based BU estimates for LULUCF presented in this study show similar magnitudes and trends compared to the NGHGIs but generally differ in the specific values. We notice stronger similarities between NGHGIs and models using national forest inventory data (e,g. CBM, EFISCEN). For cropland and grassland sector-specific models (ECOSSE, EPIC-IIASA) the differences between their results and the NGHGIs are due to differences in input

---

[18] $CO_2M$: Copernicus Anthropogenic Carbon Dioxide Monitoring,
https://esamultimedia.esa.int/docs/EarthObservation/CO2M_MRD_v3.0_20201001_Issued.pdf





data, process representation (in particular those linked to soil organic matter decomposition) and management representation. In general, management is one of the main drivers for the carbon balance of croplands and grasslands. However, spatial data on management is scarce and can have high uncertainty. For EPIC-IIASA specifically, the regional carbon simulation results for managed cropland are almost evenly impacted by model parameterization, soil input accuracy, and crop management regionalization (Balkovič et al. 2020). For the overall estimation of emissions

from LULUCF activities on all land types (Figure 9), the comparison is made more challenging as results from both land use and land use changes are presented. Comparing only the "effect of land use change" (conversion) is non-trivial and presents an area for improvement to be handled in next synthesis.

Observation-based BU estimates of LULUCF provide large year-to-year flux variability (Figures 6,7,8,9), contrary to the NGHGIs, primarily due to the effect of varying meteorology especially through the duration and

intensity of the summer growing season, which can vary significantly between years (Bastos et al., 2020; Thompson et al., 2020). In the framework of periodic NGHGI assessments, the choice of a reference period (usually five years, or a biannual reporting) may be critical in the context of large flux inter-annual variability. One direction could be to include in the NGHGIs EFs derived from the observation-based approaches (both BU and TD) in the form of year-to-year flux anomalies. The TD inversion estimates show as well pronounced inter-annual variability results (Fig. 10 and

Appendix Figure B1c for mean values). Uncertainties in the inversion results are primarily due to uncertainties in atmospheric transport modeling, boundary conditions and uncertainty inherent to the limitation of the observation network. Currently, regional inversions (CarboscopeReg and EUROCOM) are still systems under development which face different challenges from the much coarser resolution global systems used here to represent regional results (GCP ensemble incl. CarboScope global).

The next steps needed to improve and facilitate the reconciliation between BU and TD estimates will include 1) as already discussed in Petrescu et al., 2020, BU process-based models incorporating unified protocols and guidelines for uniform definitions should be able to disaggregate their estimates to facilitate comparison to NGHGI and 2006 IPCC practices (i.e., managed vs. unmanaged land, 20-year legacy for classes remaining in the same class, distinction of fluxes arising solely from land use change); 2) for sector-specific models, especially for cropland and

grassland, improving treatment of the contribution of soil organic carbon dynamic to the budget; 3) for TD estimates, using the Community Inversion Framework currently under development (Berchet et al., 2020) to better assess the different sources of uncertainties from the inversion set-ups (model transport, prior fluxes, observation networks); 4) for the overall comparison of BU and TD fluxes, incorporating the contribution of lateral fluxes of carbon by human activities and rivers that connect $CO_2$ uptake in one area with its release in another (Ciais et al., 2020).

From this analysis we demonstrate that a complete, ready-for-purpose monitoring system providing annual carbon fluxes across Europe does not yet exist. Therefore, for consistent future estimates to be used in the global stock take exercise to reach the Paris Agreement targets, significant effort must still be undertaken to reduce the uncertainty across all potential methods used in such a system (e.g. Maenhout et al., 2020).





## 5. Appendices

### *Appendix A: Data sources, methodology and uncertainty descriptions*

The country specific plots are found at: http://webportals.ipsl.jussieu.fr/VERIFY/FactSheets/ v1.24

**VERIFY project**

VERIFY's primary aim is to develop scientifically robust methods to assess the accuracy and potential biases in national inventories reported by the parties through an independent pre-operational framework. The main concept is to provide observation-based estimates of anthropogenic and natural GHG emissions and sinks as well as associated uncertainties. The proposed approach is based on the integration of atmospheric measurements, improved emission

inventories, ecosystem data, and satellite observations, and on an understanding of processes controlling GHG fluxes (ecosystem models, GHG emission models).

Two complementary approaches relying on observational data-streams will be combined in VERIFY to quantify GHG fluxes:

1) atmospheric GHG concentrations from satellites and ground-based networks (top-down atmospheric inversion

models) and

2) bottom-up activity data (e.g. fuel use and emission factors) and ecosystem measurements (bottom-up models).

For $CO_2$, a specific effort will be made to separate fossil fuel emissions from ecosystems fluxes. For $CH_4$ and $N_2O$, we will separate agricultural from fossil fuel and industrial emissions. Finally, trends in the budget of the three GHGs will be analysed in the context of NDC targets.

The objectives of VERIFY are:

**Objective 1**. Integrate the efforts between the research community, national inventory compilers, operational centres in Europe, and international organisations towards the definition of future international standards for the verification of GHG emissions and sinks based on independent observation.

**Objective 2**. Enhance the current observation and modelling ability to accurately and transparently quantify the sinks

and sources of GHGs in the land-use sector for the tracking of land-based mitigation activities.

**Objective 3.** Develop new research approaches to monitor anthropogenic GHG emissions in support of the EU commitment to reduce its GHG emissions by 40% by 2030 compared to the year 1990.

**Objective 4.** Produce periodic scientific syntheses of observation-based GHG balance of EU countries and practical policy-oriented assessments of GHG emission trends, and apply these methodologies to other countries.

For more information on project team and products/results check https://verify.lsce.ipsl.fr/.




Table A: *Country grouping used for comparison purposes between BU and TD emissions.*

| Country name – geographical Europe | BU-ISO3 | Aggregation from TD-ISO3 |
|---|---|---|
| Luxembourg | LUX | |
| Belgium | BEL | BENELUX |
| Netherlands | NLD | BNL |
| Bulgaria | BGR | BGR |
| Switzerland | CHE | |
| *Lichtenstein* | *LIE* | *CHL* |
| Czech Republic | CZE | Former Czechoslovakia |
| Slovakia | SVK | CSK |
| Austria | AUT | AUT |
| Slovenia | SVN | North Adriatic countries |
| Croatia | HRV | NAC |
| Romania | ROU | ROU |
| Hungary | HUN | HUN |
| Estonia | EST | |
| Lithuania | LTU | Baltic countries |
| Latvia | LVA | BLT |
| Norway | NOR | NOR |
| Denmark | DNK | |
| Sweden | SWE | |
| Finland | FIN | DSF |
| Iceland | ISL | ISL |
| Malta | MLT | MLT |
| Cyprus | CYP | CYP |
| France (Corsica incl.) | FRA | FRA |
| *Monaco* | *MCO* | |
| *Andorra* | *AND* | |
| Italy (Sardinia, Vatican incl.) | ITA | ITA |
| *San Marino* | *SMR* | |



| | | |
|---|---|---|
| United Kingdom (Great Britain + N Ireland) | GBR | UK |
| *Isle of Man* | *IMN* | |
| Iceland | | |
| Ireland | IRL | IRL |
| Germany | DEU | DEU |
| Spain | ESP | IBERIA |
| Portugal | PRT | IBE |
| Greece | GRC | GRC |
| *Russia (European part)* | *RUS European* | |
| *Georgia* | *GEO* | *RUS European+GEO* |
| *Russian Federation* | *RUS* | *RUS* |
| Poland | POL | POL |
| *Turkey* | *TUR* | *TUR* |
| EU27+UK (Austria, Belgium, Bulgaria, Cyprus, Czech Republic, Germany, Denmark, Spain, Estonia, Finland, France, Greece, Croatia, Hungary, Ireland, Italy, Lithuania, Latvia, Luxembourg, Malta, Netherlands, Poland, Portugal, Romania, Slovakia, Slovenia, Sweden, United Kingdom) | AUT, BEL, BGR, CYP, CZE, DEU, DNK, ESP, EST, FIN, FRA, GRC, HRV, HUN, IRL. ITA, LTU, LVA, LUX, MLT, NDL, POL, PRT, ROU, SVN, SVK, SWE, GBR | E28 |
| Western Europe (Belgium, France, United Kingdom, Ireland, Luxembourg, Netherlands) | BEL, FRA, UK, IRL, LUX, NDL | WEE |
| Central Europe (Austria, Switzerland, Czech Republic, Germany, Hungary, Poland, Slovakia) | AUT, CHE, CZE, DEU, HUN, POL, SVK | CEE |
| Northern Europe (Denmark, Estonia, Finland, Lithuania, Latvia, Norway, Sweden) | DNK, EST, FIN, LTU, LVA, NOR, SWE | NOE |
| *South-Western Europe (Spain, Italy, Malta, Portugal)* | *ESP, ITA, MLT, PRT* | *SWN* |
| *South-Eastern Europe (all) (Albania, Bulgaria, Bosnia and Herzegovina, Cyprus, Georgia, Greece, Croatia, Macedonia, the former Yugoslav, Montenegro, Romania, Serbia, Slovenia, Turkey)* | *ALB, BGR, BIH, CYP, GEO, GRC, HRV, MKD, MNE, ROU, SRB, SVN, TUR* | *SEE* |
| *South-Eastern Europe (non-EU) (Albania, Bosnia and Herzegovina, Macedonia, the former Yugoslav, Georgia, Turkey, Montenegro, Serbia)* | *ALB, BIH, MKD, MNE, SRB, GEO, TUR* | *SEA* |





| | | |
|---|---|---|
| *South-Eastern Europe (EU) (Bulgaria, Cyprus, Greece, Croatia, Romania, Slovenia)* | *BGR, CYP, GRC, HRV, ROU, SVN* | *SEZ* |
| *Southern Europe (all) (SOE) (Albania, Bulgaria, Bosnia and Herzegovina, Cyprus, Georgia, Greece, Croatia, Macedonia, the former Yugoslav, Montenegro, Romania, Serbia, Slovenia, Turkey, Italy, Malta, Portugal, Spain)* | *ALB, BGR, BIH, CYP, GEO, GRC, HRV, MKD, MNE, ROU, SRB, SVN, TUR, ITA, MLT, PRT, ESP* | *SOE* |
| *Southern Europe (non-EU) (SOY) Albania, Bosnia and Herzegovina, Georgia, Macedonia, the former Yugoslav, Montenegro, Serbia, Turkey)* | *ALB, BIH, GEO, MKD, MNE, SRB, TUR,* | *SOY* |
| Southern Europe (EU) (SOZ) (Bulgaria, Cyprus, Greece, Croatia, Romania, Slovenia, Italy, Malta, Portugal, Spain) | BGR, CYP, GRC, HRV, ROU, SVN, ITA, MLT, PRT, ESP | SOZ |
| Eastern Europe (non-EU) (Belarus, Moldova, Republic of, *Russian Federation*, Ukraine) | BLR, MDA, *RUS*, UKR | EAE |
| *EU-15 (Austria, Belgium, Germany, Denmark, Spain, Finland, France, United Kingdom, Greece, Ireland, Italy, Luxembourg, Netherlands, Portugal, Sweden)* | *AUT, BEL, DEU, DNK, ESP, FIN, FRA, GBR, GRC, IRL, ITA, LUX, NDL, PRT, SWE* | *E15* |
| *EU-27 (Austria, Belgium, Bulgaria, Cyprus, Czech Republic, Germany, Denmark, Spain, Estonia, Finland, France, Greece, Croatia, Hungary, Ireland, Italy, Lithuania, Latvia, Luxembourg, Malta, Netherlands, Poland, Portugal, Romania, Slovakia, Slovenia, Sweden)* | *AUT, BEL, BGR, CYP, CZE, DEU, DNK, ESP, EST, FIN, FRA, GRC, HRV, HUN, IRL. ITA, LTU, LVA, LUX, MLT, NDL, POL, PRT, ROU, SVN, SVK, SWE* | *E27* |
| *All Europe (Aaland Islands, Albania, Andorra, Austria, Belgium, Bulgaria, Bosnia and Herzegovina, Belarus, Switzerland, Cyprus, Czech Republic, Germany, Denmark, Spain, Estonia, Finland, France, Faroe Islands, United Kingdom, Guernsey, Greece, Croatia, Hungary, Isle of Man, Ireland, Iceland, Italy, Jersey, Liechtenstein, Lithuania, Luxembourg, Latvia, Moldova, Republic of, Macedonia, the former Yugoslav, Malta, Montenegro, Netherlands, Norway, Poland, Portugal, Romania, Russian Federation, Svalbard and Jan Mayen, San Marino, Serbia, Slovakia, Slovenia, Sweden, Turkey, Ukraine)* | *ALA, ALB, AND, AUT, BEL, BGR, BIH, BLR, CHE, CYP, CZE, DEU, DNK, ESP, EST, FIN, FRA, FRO, GBR, GGY, GRC, HRV, HUN, IMN, IRL, ISL, ITA, JEY, LIE, LTU, LUX, LVA, MDA, MKD, MLT, MNE, NDL, NOR, POL, PRT, ROU, RUS, SJM, SMR, SRB, SVK, SVN, SWE, TUR, UKR* | *EUR* |

*countries highlighted in *italic* are not discussed in the current 2019 synthesis mostly because unavailability of UNFCCC NGHGI reports (non-Annex I countries[19]) but are present on the web-portal: http://webportals.ipsl.jussieu.fr/VERIFY/FactSheets/. Results of Annex I countries (NOR, CHE, ISL) and non-EU Eastern European countries (EAE) are represented in Figure 4.

---

[19]Non-Annex I countries are mostly developing countries. The reporting to UNFCCC is implemented through national communications (NCs) and biennial update reports (BURs): https://unfccc.int/national-reports-from-non-annex-i-parties






*Table AA: Methodological changes (**in bold**) of current study with respect to Petrescu et al., 2020; n/a cells mean that there is no data available.*

| Publication year | Bottom-up anthropogenic CO$_2$ estimates (fossil CO$_2$) | | | Top-down fossil CO$_2$ estimates | Bottom-up natural CO$_2$ (NBP) emissions/removals (land CO$_2$) | | | Top-down land CO$_2$ emissions | | Uncertainty and other changes |
|---|---|---|---|---|---|---|---|---|---|---|
| | Inventories | Global databases | Emission models | | Inventories | Emission models | Global Databases | Regional models | Global models | |
| 2020<br><br>Petrescu et al. (2020) AFOLU bottom-up synthesis | n/a | n/a | n/a | n/a | National emissions from UNFCCC (2018) 1990-2016<br><br>*LULUCF Forest land, -* EU28 data for five years (1995, 2000, 2005, 2010 and 2015) *Cropland and Grassland* (1990, 2005, 2010 and 2016) *All land uses* EU28 time series 1990-2016 | CBM Forest land (2000, 2005, 2010 and 2015)<br><br>EFISCEN Forest Land (1995, 2000, 2005, 2010 and 2015)<br><br>BLUE All land uses 1990-2017<br><br>H&N All land uses 1990-2015<br><br>DGVMs (TRENDY v6) All land uses 1990-2017 | FAOSTAT Time series Remaining and conversions 1990-2016 | n/a | n/a | UNFCCC (2018) uncertainty estimates for 2016 (error propagation 95% interval method) |
| **2021**<br><br>this study synthesis bottom-up **and top-down** | National emissions from UNFCCC **(2019)** CRFs **2014**<br><br>**All anthropogenic (excl. LULUCF) sectors, time series** 1990-2015 | **EDGAR v5.0 BP EIA CDIAC IEA GCP CEDS**<br><br>**2014 estimates split by fuel type**<br><br>**EDGAR v5.0 All anthropogenic sectors, time series** | n/a | **IAP RAS fast-track inversion 2014 (EU11+CHE)** | National emissions from UNFCCC **(2019)** 1990-**2017 EU27 + UK Time series of Forest Land, Cropland and Grassland**<br><br>**Regional EU27 + UK totals (incl. NOR, CHE, UKR, MLD and BLR)** | CBM Forest land **time series 1990-2015**<br><br>EFISCEN Forest Land **Time series 2005-2018**<br><br>**CO$_2$ emissions from inland waters**<br><br>**ORCHIDEE Forest, cropland and grassland and all land uses 1990-2018** | FAOSTAT Time series Remaining and conversions 1990-**2017** | **CarboScopeReg 2006-2018**<br><br>**EUROCOM 2006-2015** | **GCP 2019 inversions 2000-2018** | UNFCCC **(2019)** uncertainty estimates for 2016 (error propagation 95% interval method)<br><br>**For model ensembles reported as variability in extremes (min/max)** |





| | | 1990-2015 | | | | ECOSSE Cropland and grassland 1990-2018 | | | | |
|---|---|---|---|---|---|---|---|---|---|---|
| | | | | | | EPIC-IIASA Cropland 1990-2018 | | | | |
| | | | | | | BLUE All land uses 1990-**2018** | | | | |
| | | | | | | H&N All land uses 1990-2015 | | | | |
| | | | | | | DGVMs (TRENDY v7) All land uses 1990-**2018** | | | | |

**A1: Fossil CO₂ emissions**

***Bottom-up emission estimates***

***For further details, see Andrew (2020)***

***UNFCCC NGHGI (2019)***

865       The Annex-I parties to the UNFCCC are required to report emissions inventories annually using the Common Reporting Format (CRF). This annual published dataset includes all $CO_2$ emissions sources for those countries, and for most countries for the period 1990 to t-2. Some eastern European countries' submissions begin in the 1980s. Revisions are made on an irregular basis outside of the standard annual schedule. For complete description see Andrew, 2020.

**Uncertainties:** Annex I parties quantitatively estimate their uncertainties of data used for all source and sink categories using the methodologies provided in the 2006 IPCC Guidelines. The 2006 IPCC Guidelines stipulate that the determination of uncertainties is a key element of any complete inventory. Uncertainties are quantified for emission factors, activity data and, in some cases, for emissions. In general, two methods for determining uncertainties are differentiated. The Tier 1 method combines, in a simple way, the uncertainties in activity data and emission factors,

for each category and greenhouse gas, and then aggregates these uncertainties, for all categories and greenhouse gas components, to obtain the total uncertainty for the inventory. The Tier 2 method for uncertainties determination is the same, in principle, but it also considers the distribution function for uncertainties and carries out aggregation using Monte Carlo simulation. In the Tier 2 method, the process also necessarily includes the determination of the probability density function for both parameters.





### EDGAR v5.0

The first edition of the Emissions Database for Global Atmospheric Research was published in 1995. The dataset now includes almost all sources of fossil $CO_2$ emissions, is updated annually, and reports data for 1970 to n-1. Estimates are provided by sector. Emissions are estimated fully based on statistical data from 1970 till 2015, while for the years beyond 2015 a Fast Track (FT) approach is applied based on BP data and other proxies to extrapolate $CO_2$ emissions till most recent years (Crippa et al., 2019b, https://edgar.jrc.ec.europa.eu/overview.php?v=booklet2019). For complete description see Andrew, 2020.

**Uncertainties:** EDGAR uses emission factors (EFs) and activity data (AD) to estimate emissions. Both EFs and AD are uncertain to some degree, and when combined, their uncertainties need to be combined too. To estimate EDGAR's uncertainties (stemming from lack of knowledge of the true value of the EF and AD), the methodology devised by IPCC (2006, Chapter 3) is adopted, that is the sum of squares of the uncertainty of the EF and AD (uncertainty of the product of two variables). A log-normal probability distribution function is assumed to avoid negative values, and uncertainties are reported as 95 % confidence interval according to IPCC (2006, chapter 3, equation 3.7). For emission uncertainty in the range 50 % to 230% a correction factor is adopted as suggested by Frey et al (2003) and IPCC (2006, chapter 3, equation 3.4).

### BP

BP releases its Statistical Review of World Energy annually in June, the first report being published in 1952. Primarily an energy dataset, BP also includes estimates of fossil-fuel $CO_2$ emissions derived from its energy data. The emissions estimates are totals for each country starting in 1965 to year n-1. For complete description see Andrew, 2020.

### CDIAC

The original Carbon Dioxide Information Analysis Center included a fossil $CO_2$ emissions dataset that was long known as CDIAC. This dataset is now produced at Appalachian State University, and includes emissions from fossil fuels and cement production from 1751 to n-3. Fossil-fuel emissions are derived from UN energy statistics, and cement emissions from USGS production data. For complete description see Andrew, 2020.

### EIA

The US Energy Information Administration publishes international energy statistics and from these derives estimates of energy combustion $CO_2$ emissions. Data are currently available for the period 1980-2016. For complete description see Andrew, 2020.

### IEA

The International Energy Agency publishes international energy statistics and from these derives estimates of energy combustion $CO_2$ emissions including from the use of coal in the iron and steel industry. Emissions estimates start in 1960 for OECD members and 1971 for non-members, and run through n-1 for OECD members' totals, and n-2 for members' details and non-members. Estimates are available by sector for a fee. For complete description see Andrew, 2020.





### GCP

The Global Carbon Project includes estimates of fossil $CO_2$ emissions in its annual Global Carbon Budget publication. These includes emissions from fossil fuels and cement production for the period 1750 to n-1. For complete description see Andrew, 2020.

### CEDS

The Community Emissions Data System has included estimates of fossil $CO_2$ emissions since 2018, with an irregular update cycle. Energy data are directly from IEA, but emissions are scaled to higher-priority sources, including national inventories. Almost all emissions sources are included and estimates are published for the period 1750-2014. Estimates are provided by sector. For complete description see Andrew R. M., ESSD, 2020.


### Top-down $CO_2$ emission estimates

### Fast-track fossil $CO_2$ emission inversion

The so called KL18 inversion product (Konovalov and Lvova 2018) consists in a rescaling of the 0.1° x 0.1° resolution maps of annual averages of fossil $CO_2$ anthropogenic emissions in Western Europe (over 11 countries of
the European Union -Portugal, Spain, France, Belgium, Luxembourg, Netherlands, UK, Germany, Denmark, Italy, Austria- and Switzerland) from the EDGARv4.3.2 inventory (Janssens et al., 2019). It has been produced by IAP-RAS to provide first inversions of the emissions in Europe during the first years of VERIFY, while the development of the main inversion system for this task should last more than 2 years. It covers the years 2012 to 2015, updating the method and extending the inversions documented in Konovalov et al. (2016). The factors scaling the EDGARv4.3.2
maps are derived from the regional inversions of CO and NOx emissions using EMEP/CEIP as prior knowledge of the emissions and $CO_2$/CO and $CO_2$/NOx emission ratios associated with the combustion of fossil from EDGARv4.3.2. These regional inversions are based on the assimilation of satellite atmospheric concentration data: total column CO from IASI, and tropospheric column $NO_2$ from OMI in a 50-km resolution European configuration of the CHIMERE mesoscale atmospheric chemistry transport model (Menut et al., 2013). The resulting
$fCO_2$ inverse emissions are calculated by converting the inverted CO and NOx emission (sectoral or total) budgets into $fCO_2$ emissions budgets using ratios of CO (all emissions) / $fCO_2$ (fossil fuel missions only) and NOx (all emissions) / $fCO_2$ (fossil fuel emissions only) from EDGAR (excluding biofuel from the $CO_2$ emissions in EDGAR but not from the CO and NOx emissions in EDGAR).

**Uncertainty:** An estimate of the uncertainty in the annual budgets of the emissions over the 12 countries is derived
from the analyzes of uncertainties within the CO and NOx emission inversions (associated with model and data errors) and from an assessment of the uncertainties in the $CO_2$/CO and $CO_2$/NOx emission ratios (based on their spatial variability). The preliminary results indicate that the uncertainty in the information from the CO inversion is too high to provide reliable estimates of the $CO_2$ fossil emissions when using CO satellite data only, or to provide weight to this information when using $CO_2$ fossil estimates from both the CO and $NO_x$ inversions. The estimates based on
$NO_2$ data are close to EDGAR v4.3.2 in 2012. These estimates are quite constant over the 4-year period while we



assume that the $CO_2$ fossil emissions followed a significant negative trend during this period. The analysis shows that the uncertainties in these estimates can explain the difficulty to detect such a trend.

**A2: Land $CO_2$ emissions/removals**

*Bottom-up $CO_2$ estimates*

*UNFCCC NGHGI 2019 - LULUCF*

Under the convention and its Kyoto Protocol national greenhouse gas (GHG) inventories are the most important source of information to track progress and assess climate protection measures by countries. In order to build mutual trust in the reliability of GHG emission information provided, national GHG inventories are subject to
standardized reporting requirements, which have been continuously developed by the Conference of the Parties (COP)[20]. The calculation methods for the estimation of greenhouse gases in the respective sectors is determined by the methods provided by the 2006 IPCC Guidelines for National Greenhouse Gas Inventories (IPCC, 2006). They provide detailed methodological descriptions to estimate emissions and removals, as well as provide recommendations to collect the activity data needed. As a general overall requirement, the UNFCCC reporting guidelines stipulate that
reporting under the Convention and the Kyoto Protocol must follow the five key principles of transparency, accuracy, completeness, consistency and comparability (TACCC). The reporting under UNFCCC shall meet the TACCC principles. The three main GHGs are reported in time series from 1990 up to two years before the due date of the reporting. The reporting is strictly source category based and is done under the Common Reporting Format tables (CRF), downloadable from the UNFCCC official submission portal:
https://unfccc.int/process-and-meetings/transparency-and-reporting/reporting-and-review-under-the-convention/greenhouse-gas-inventories-annex-i-parties/national-inventory-submissions-2019

For the biogenic $CO_2$ emissions from sector 4 LULUCF, methods for the estimation of $CO_2$ removals and differ enormously among countries and land use categories. Each country uses its own country specific method which takes into account specific national circumstances (as long as they are in accordance with the 2006 IPCC guidelines),
as well as IPCC default values, which are usually more conservative and result in higher uncertainties. The EU GHG inventory underlies the assumption that the individual use of national country specific methods leads to more accurate GHG estimates than the implementation of a single EU wide approach (UNFCCC, 2018b). Key categories for the EU28 are 4.A.1 Forest Land: Land Use $CO_2$, 4.A.2. Forest Land: Land Use $CO_2$, 4.B.1 Cropland Land Use $CO_2$, 4.B.2 Cropland Land Use $CO_2$, 4.C.1 Grassland Land Use $CO_2$, 4.C.2 Grassland Land Use $CO_2$, 4.D.1 Wetlands Land Use
$CO_2$, 4.E.2 Settlements Land Use $CO_2$, and 4.G Harvested Wood Production Wood product $CO_2$. The tier method a country applies depends on the national circumstances and the individual conditions of the land, which explains the variability of uncertainties among the sector itself as well as among EU countries.

---

[20]The last revision has been made by COP 19 in 2013 (UNFCCC, 2013)





**Uncertainty** methodology for the NGHGI UNFCCC submissions are based on the Chapter 3 of 2006 IPCC Guidelines for National Greenhouse Gas Inventories and is the same as in paragraph 2.1 and ESSDD Petrescu et al., 2019, Appendix B.

### *ORCHIDEE*

ORCHIDEE is a general ecosystem model designed to be coupled to an atmospheric model in the context of modeling the entire Earth system. As such, ORCHIDEE calculates its prognostic variables (i.e., a multitude of C, $H_2O$ and energy fluxes) from the following environmental drivers: air temperature, wind speed, solar radiation, air humidity, precipitation and atmospheric $CO_2$ concentration. As the run progresses, vegetation grows on each pixel, divided into thirteen generic types (e.g., broadleaf temperate forests, C3 crops), which cycle carbon between the soil, land surface, and atmosphere, through such processes such as photosynthesis, litter fall, and decay. Limited human activities are included through the form of generic wood and crop harvests, which remove aboveground biomass on an annual basis.

Among other environmental indicators, ORCHIDEE simulates positive and negative $CO_2$ emissions from plant uptake, soil decomposition, and harvests across forests, grasslands, and croplands. Activity data is based on land use and land cover maps. For VERIFY, pixel land cover/land use fractions were based on the land use map LUH2v2h and the land cover project of the Climate Change Initiative (CCI) program of the European Space Agency (ESA). The latter is based on purely remotely-sensed methods, while the former makes use of national harvest data from the U.N. Food and Agricultural Organization.

**LUH2v2-ESACCI**: "We describe here the input data and algorithms used to create the land cover maps specific for our CMIP6 simulations using the historical/future reconstruction of land use states provided as reference datasets for CMIP6 within the land use harmonization database LUH2v2h (Hurtt et al., 2011). More details are provided on the devoted web page https://orchidas.lsce.ipsl.fr/dev/lccci which shows further tabular, graphical and statistical data. The overall approach relies on the combination of the LUH2v2 data with present-day land cover distribution derived from satellite observations for the past decades. The main task consists in allocating the land-use types from LUH2v2 in the different PFTs for the historical period and the future scenarios. The natural vegetation in each grid cell is defined as the PFT distribution derived from the ESA-CCI land cover product for the year 2010 to which pasture fraction and crop fraction from LUH2v2 (for the year 2010) have been subtracted from grass and crop PFTs. This characterization of the natural vegetation in terms of PFT distribution is assumed invariant in time and is used for both the historical period and the different future scenarios." (Lurton et al., 2020).

**Uncertainty** in the ORCHIDEE model arises from three primary sources: parameters, forcing data (including spatial and temporal resolution), and model structure. Some researchers argue that the initial state of the model (i.e., the values of the various carbon and water pools at the beginning of the production run, following model spinup) represents a fourth area. However, the initial state of the model is defined by its equilibrium state, and therefore a strong function of the parameters, forcing data, and model structure, with the only independent choice being the target year of the initial state. Out of the three primarily areas of uncertainty, the climate forcing data is dictated by the VERIFY project itself, thus removing that source from explaining observed differences among the models, although



it can still contribute to uncertainty between the ORCHIDEE results and the national inventories. The land use/land cover maps, another major source of uncertainty for ORCHIDEE carbon fluxes, have also been harmonized to a large extent between the bottom-up carbon budget models in the project. Parameter uncertainty and model structure thus represent the two largest sources of potential disagreement between ORCHIDEE and the other bottom-up carbon budget models. Computational cost prevents a full characterization of uncertainty due to parameter selection in

ORCHIDEE (and dynamic global vegetation models in general), and uncertainties in model structure require the use of multiple models of the same type but including different physical processes. Such a comparison has not been done in the context of VERIFY, although the results from the TRENDY suite of models shown in Figures (plots Matt BU-TD) give a good indication of this."

*$CO_2$ Emissions from inland waters*

These estimates represent a climatology of average annual $CO_2$ emissions from rivers, lakes and reservoirs at the spatial resolution of 0.1°. The approach combines $CO_2$ evasion fluxes from the global river network, as estimated by the empirical model of Lauerwald et al. (2015) with the lakes and reservoirs estimates by Hastie et al. (2019) for the boreal biome and by Raymond et al. (2013) for the lower latitudes. The Lauerwald et al. and Hastie et al. studies

follow the same approach and rely on the development of a statistical prediction model for inland water $pCO_2$ at 0.5° using global, high-resolution geodata. The $pCO_2$ climatology was then combined with different estimates of the gas transfer velocity k to produce the resulting map of $CO_2$ evasion. The Raymond et al. study only provides mean flux densities at the much coarser spatial resolution of the so-called COSCAT regions. All estimates were then downscaled to 0.1° using the spatial distribution of European inland water bodies. Note that in contrast to Hastie et al. (2019), the

areal distribution of lakes was extracted from the HYDROLAKES database (Messager et al., 2016), to be consistent with the estimates of inland water $N_2O$ and $CH_4$ presented by Petrescu et al., 2020 in review at ESSD.

**Uncertainty**: Monte Carlo simulations were performed to constrain uncertainties resulting from both the $pCO_2$ prediction equation and the choice of the k formulation.

*CBM*

The Carbon Budget Model developed by the Canadian Forest Service (CBM-CFS3), can simulate the historical and future stand- and landscape-level C dynamics under different scenarios of harvest and natural disturbances (fires, storms), according to the standards described by the IPCC (Kurz et al., 2009). Since 2009, the CBM has been tested and validated by the Joint Research Centre of the European Commission (JRC), and adapted to

the European forests. It is currently applied to 26 EU Member States, both at country and NUTS2 level (Pilli et al., 2016b).

Based on the model framework, each stand is described by area, age and land use classes and up to 10 classifiers based on administrative and ecological information and on silvicultural parameters (such as forest composition and management strategy). A set of yield tables define the merchantable volume production for each

species while species-specific allometric equations convert merchantable volume production into aboveground





biomass at stand-level. At the end of each year the model provides data on the net primary production (NPP), carbon stocks and fluxes, as the annual C transfers between pools and to the forest product sector.

The model can support policy anticipation, formulation and evaluation under the LULUCF sector, and it is used to estimate the current and future forest C dynamics, both as a verification tool (i.e., to compare the results with the estimates provided by other models) and to support the EU legislation on the LULUCF sector (Grassi et al., 2018a). In the biomass sector, the CBM can be used in combination with other models, to estimate the maximum wood potential and the forest C dynamic under different assumptions of harvest and land use change (Jonsson et al., 2018). Uncertainty: Quantifying the overall uncertainty of CBM estimates is challenging because of the complexity of each parameter. The uncertainty in CBM arises from three primary sources: parameters, forcing data (including spatial and temporal resolution) and model structure. It is linked to both activity data and emission factors (area, biomass volume implied by species specific equation to convert the merchantable volume to total aboveground biomass (used as a biomass expansion factor)) as well to the capacity of each model to represent the original values, in this case estimated through the mean percentage difference between the predicted and observed values. A detailed description of the uncertainty methodology is found in Pilli et al., 2017.

### EFISCEN

The European Forest Information SCENario Model (EFISCEN) is a large-scale forest model that projects forest resource development on regional to European scale. The model uses national forest inventory data as a main source of input to describe the current structure and composition of European forest resources. The model projects the development of forest resources, based on scenarios for policy, management strategies and climate change impacts. With the help of biomass expansion factors, stem wood volume is converted into whole-tree biomass and subsequently to whole tree carbon stocks. Information on litter fall rates, felling residues and natural mortality is used as input into the soil module YASSO (Liski et al. 2005), which is dynamically linked to EFISCEN and delivers information on forest soil carbon stocks. The core of the EFISCEN model was developed by Prof. Ola Sallnäs at the Swedish Agricultural University (Sallnäs 1990). It has been applied to European countries in many studies since then, dealing with a diversity of forest resource and policy aspects. A detailed model description is given by Verkerk et al. (2016), with online information on availability and documentation of EFISCEN at http://efiscen.efi.int. The model and its source code are freely available, distributed under the GNU General Public License conditions (www.gnu.org/licenses/gpl-3.0.html).

**Uncertainties:** Sensitivity analysis on EFISCEN v3 is described in detail by Schelhaas et al. 2007 (the manual). Total sensitivity is caused by especially young forest growth, width of volume classes, age of felling and few more. Scenario uncertainty comes on top of this when projecting in future.

### EPIC-IIASA (croplands)

The Environmental Policy Integrated Climate (EPIC) model is a field-scale process-based model (Izaurralde et al., 2006; Williams, 1990) which calculates, with a daily time step, crop growth and yield, hydrological,



nutrient and carbon cycling, soil temperature and moisture, soil erosion, tillage, and plant environment control. Potential crop biomass is calculated from photosynthetically active radiation using the radiation-use efficiency concept modified for vapor pressure deficit and atmospheric $CO_2$ concentration effect. Potential biomass is adjusted to actual

biomass through daily stress caused by extreme temperatures, water and nutrient deficiency or inadequate aeration. The coupled organic C and N module in EPIC (Izaurralde et al., 2006) distributes organic C and N between three pools of soil organic matter (active, slow, and passive) and two litter compartments (metabolic and structural). EPIC calculates potential transformations of the five compartments as regulated by soil moisture, temperature, oxygen, tillage and lignin content. Daily potential transformations are adjusted to actual transformations when the combined

N demand in all receiving compartments exceeds the N supply from the soil. The transformed components are partitioned into $CO_2$ (heterotrophic respiration), dissolved C in leaching (DOC), and the receiving SOC pools. EPIC also calculates SOC loss with erosion.

The EPIC-IIASA (version EU) modelling platform was built by coupling the field-scale EPIC v. 0810 with large-scale data on land cover (cropland), soils, topography, field size, and crop management practices aggregated at

a 1×1 km grid covering European countries (Balkovič et al., 2018, 2013). In VERIFY, a total of ten major European crops including winter wheat, winter rye, spring barley, grain maize, winter rapeseed, sunflower, sugar beet, potatoes, soybean, and rice were used to represent agricultural production systems in Europe. Crop fertilization and irrigation were estimated for NUTS2 statistical regions between 1995 and 2010 (Balkovič et al., 2013). For VERIFY, the simulations were carried out assuming conventional tillage, consisting of two cultivation operations and mouldboard

ploughing prior to sowing and an offset disking after harvesting of cereals. Two row cultivations during the growing season were simulated for maize and one ridging operation for potatoes. It was assumed that 20% of crop residues are removed in case of cereals (excluding maize), while no residues are harvested for other crops.

**Uncertainties** in EPIC arise from three primary sources which were in detail described by ORCHIDEE. A detailed sensitivity and uncertainty analysis of EPIC-IIASA regional carbon modelling is presented in Balkovič et al. (2020).


### *ECOSSE (grasslands)*

ECOSSE is a biogeochemical model that is based on the carbon model ROTH-C (Jenkinson and Rayner, 1977; Jenkinson et al. 1987; Coleman and Jenkinson, 1996) and the nitrogen-model SUNDIAL (Bradbury et al. 1993; Smith et al. 1996). All processes of the carbon and nitrogen dynamics are considered (Smith et al., 2010a,b).

Additionally, in ECOSSE processes of minor relevance for mineral arable soils are implemented as well (e.g. methane emissions) to have a better representation of processes that are relevant for other soils (e.g. organic soils). ECOSSE can run in different modes and for different time steps. The two main modes are site specific and limited data. In the later version, basis assumptions/estimates for parameters can be provided by the model. This increases the uncertainty but makes ECOSSE a universal tool that can be applied for largescale simulations even if the data availability is

limited. To increase the accuracy in the site-specific version of the model, detailed information about soil properties, plant input, nutrient application and management can be added as available.

During the decomposition process, material is exchanged between the SOM pools according to first order rate equations, characterised by a specific rate constant for each pool, and modified according to rate modifiers





dependent on the temperature, moisture, crop cover and pH of the soil. The model includes five pools with one of them are inert. The N content of the soil follows the decomposition of the SOM, with a stable C:N ratio defined for each pool at a given pH, and N being either mineralised or immobilised to maintain that ratio. Nitrogen released from decomposing SOM as ammonium (NH4+) or added to the soil may be nitrified to nitrate (NO3-).

For spatial simulations the model is implemented in a spatial model platform. This allows to aggregate the input parameter for the needed resolution. ECOSSE is a one-dimensional model and the model platform provides the input data in a spatial distribution and aggregates the model outputs for further analysis. While climate data are interpolated, soil data are represented by the dominant soil type or by the proportional representation of the different soil types in the spatial simulation unit (this is in VERIFY a grid cell).

**Uncertainties** in ECOSSE arise from three primary sources: parameters, forcing data (including spatial and temporal resolution), and model structure. These uncertainties are not yet quantified.

### *Bookkeeping models*

We make use of data from two bookkeeping models: **BLUE** (Hansis et al., 2015) and **H&N** (Houghton & Nassikas, 2017).

The **BLUE** model provides a data-driven estimate of the net land use change fluxes. BLUE stands for 'bookkeeping of land use emissions'. Bookkeeping models (Hansis 2015, Houghton 1983) calculate land-use change $CO_2$ emissions (sources and sinks) for transitions between various natural vegetation types and agricultural lands. The bookkeeping approaches keep track of the carbon stored in vegetation, soils, and products before and after the land-use change.

In BLUE, land-use forcing is taken from the Land Use Harmonization, LUH2, for estimates within the annual global carbon budget. The model provides data at annual time steps and 0.25 degree resolution. Temporal evolution of carbon gain or loss, i.e., how fast carbon pools decay or regrow following a land-use change, is based on response curves derived from literature. The response curves describe decay of vegetation and soil carbon, including transfer to product pools of different lifetimes, as well as carbon uptake due to regrowth of vegetation and subsequent refilling of soil carbon pools.

**Uncertainties** are not explicitly quantified in BLUE so far. A large contribution of uncertainty can be expected from various input datasets. Apparent uncertainties arise from the land-use forcing data, the equilibrium carbon densities of soil and vegetation and the response curves build to reflect carbon pools decay and regrow after land-use transitions. Furthermore, Hansis et al 2015 have shown that different accounting schemes and initialization settings lead to different emission estimates even decades after the model start.

The **H&N** model (Houghton, 1983) calculates land-use change $CO_2$ emissions and uptake fluxes for transitions between various natural vegetation types and agricultural lands (croplands and pastures). The original bookkeeping approach of Houghton (2003) keeps track of the carbon stored in vegetation and soils before and after the land-use change. Carbon gain or loss is based on response curves derived from literature. The response curves describe decay of vegetation and soil carbon, including transfer to product pools of different life-times, as well as carbon uptake due to regrowth of vegetation and consequent re-filling of soil carbon pools. Natural vegetation can



generally be distinguished into primary and secondary land. For forests, a primary forest that is cleared cannot recover back to its original carbon density. Instead, long- term degradation of primary forest is assumed and represented by lowered standing vegetation and soil carbon stocks in the secondary forests. Apart from land use transitions between different types of vegetation cover, forest management practices in the form of wood harvest volumes are included.

Different from dynamic global vegetation models, bookkeeping models ignore changes in environmental conditions (climate, atmospheric $CO_2$, nitrogen deposition and other environmental factors). Carbon densities at a given point in time are only influenced by the land use history, but not by the preceding changes in the environmental state. Carbon densities are taken from observations in the literature and thus reflect environmental conditions of the last decades.

*FAOSTAT*

FAOSTAT: Statistics Division of the Food and Agricultural Organization of the United Nations provides LULUCF $CO_2$ emissions for the period 1990-2017, available at: http://www.fao.org/faostat/en/#data/GL and its sub-domains. The FAOSTAT emissions land use database (metadata: http://fenixservices.fao.org/faostat/static/documents/GL/GL_e_2019.pdf ) is computed following Tier 1 IPCC 2006

Guidelines for National GHG Inventories (http://www.ipcc-nggip.iges.or.jp/public/2006gl/index.html). Geospatial data are the source of AD for the estimates of emissions from cultivation of organic soils, biomass and peat fires. GHG emissions are provided by country, regions and special groups, with global coverage, relative to the period 1990-present (with annual updates). Land Use Total contains all GHG emissions and removals produced in the different Land Use sub-domains, representing four IPCC Land Use categories, of which three land use categories: forest land,

cropland, grassland; plus and biomass burning. LULUCF emissions consist of $CO_2$ associated with land use and change, including management activities. $CO_2$ emissions/removals are computed at Tier 3 using carbon stock change. To this end, FAOSTAT uses Forest area and carbon stock data from FRA, gap-filled and interpolated to generate annual time-series. As a result $CO_2$ emissions/removals are computed for forest land and net forest conversion, representing respectively IPCC categories ''forest land'' and ''forest land converted to other land uses''. $CO_2$

emissions are provided as by country, regions and special groups, with global coverage, relative to the period 1990-most recent available year (with annual updates), expressed as net emissions/removals as Gg $CO_2$, by underlying land use emission sub-domain and by aggregate (land use total).

**Uncertainties** are not available for the FAOSTAT estimates.

*TRENDY v7*

The TRENDY (Trends in net land-atmosphere carbon exchange over the period 1980-2010) project represents a consortium of dynamic global vegetation models (DGVMs) following identical simulation protocols to investigate spatial trends in carbon fluxes across the globe over the past century. As DGVMs, the models require climate, carbon dioxide, and land use change input data to produce results. In TRENDY, all three of these are

harmonized to make the results across the whole suite of models more comparable. In the case of VERIFY, we used the following 14 models from version 7 of TRENDY, released in 2018 and therefore covering the period up to and





including 2017: CABLE, CLASS, CLM5, DLEM, ISAM, JSBACH, JULES, LPJ, LPX, OCN, ORCHIDEE-CNP, ORCHIDEE, SDGVM, SURFEX.

While describing the details of all the models used here is clearly not possible, DGVMs calculate prognostic variables (i.e., a multitude of C, $H_2O$ and energy fluxes) from the following environmental drivers: air temperature, wind speed, solar radiation, air humidity, precipitation and atmospheric $CO_2$ concentration. As the run progresses, vegetation grows on each pixel, divided into generic types which depend on the model (e.g., broadleaf temperate forests, C3 crops), which cycle carbon between the soil, land surface, and atmosphere, through such processes such as photosynthesis, litter fall, and decay. Limited human activities are included depending on the model, typically removing aboveground biomass on an annual basis.

Among other environmental indicators, DGVMs simulate positive and negative $CO_2$ emissions from plant uptake, soil decomposition, and harvests across forests, grasslands, and croplands. Activity data is based on land use and land cover maps. For TRENDY, pixel land cover/land use fractions were based on the land use map LUH2 (Hurtt et al 2011, 2020) and the HYDE land_use change data set (Klein Goldewijk et al., 2017a, b). Both of these maps rely on FAO statistics on agricultural land area and national harvest data.

**Uncertainties** in TRENDY v7 are model specific and described by Le Quéré et al., 2018. The spread of the 14 TRENDY models used by this study (Fig. 9) gives an idea of the uncertainty due to model structure in dynamic global vegetation models, as the forcing data was harmonized for all models.

### *Top-down $CO_2$ emissions estimates*

### *CarboScope-Regional, GCP 2019 (CTE, CAMS, CarboScope) and EUROCOM*

Top-down estimates of land biosphere fluxes are provided by a number of different inverse modeling systems that use atmospheric concentration data as input, as well as prior information on fossil emissions, ocean fluxes, and land biosphere fluxes. The land biosphere fluxes, and in some systems the ocean fluxes, are estimated using a statistical optimization involving atmospheric transport models. The inversion systems differ in the transport models used, optimization methods, spatiotemporal resolution, boundary conditions, and prior error structure (spatial and temporal correlation scales), thus using ensembles of such systems is expected to result in more robust top-down estimates.

For this study, the global inversion results are taken from the GCP 2019 (Global Carbon Project) models CTE (CarbonTracker Europe), CAMS (Copernicus Atmosphere Monitoring Service), and CarboScope, with spatial resolutions ranging from 1°x1° for certain regions to 4°x5°. For details see Friedlingstein et al., 2019.

Top-down estimates at regional scales (up to 0.25°x0.25° resolution) for the period 2006 – 2015 are taken from the six models used within EUROCOM (Monteil et al., 2019). These inversions make use of more than 30 atmospheric observing stations within Europe, including flask data and continuous observations. The CarboScope-Regional (CSR) inversion system (also included within the EUROCOM ensemble) was also run for the extended period 2006- 2018 using four different settings: three network configurations using 15, 40, or 46 sites, and one using all 46 sites but a factor two larger prior error correlation length scale (200 instead of 100 km).



***Appendix B***

***B1: Overview figures***

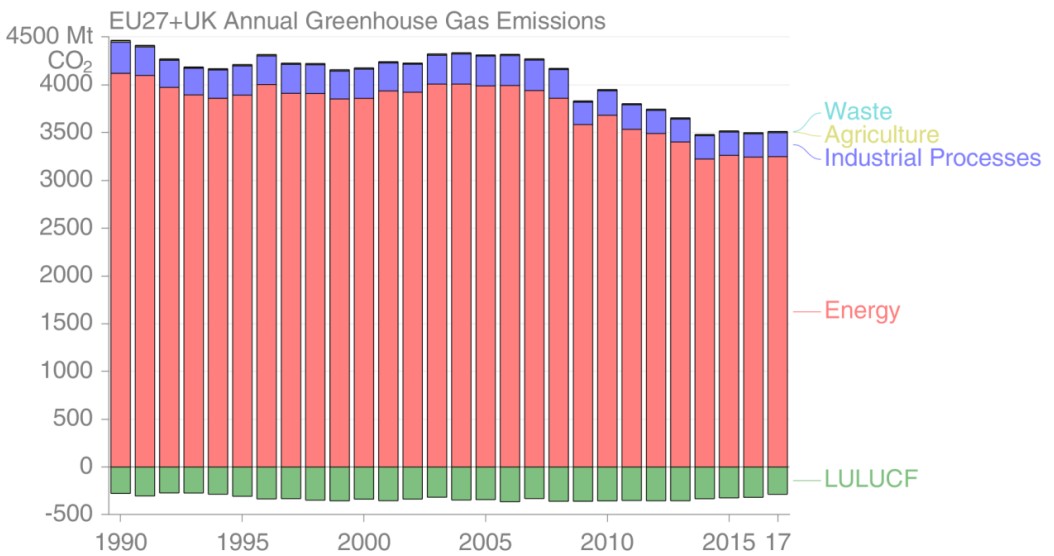


*Figure B1a: EU27+UK total annual GHG emissions from UNFCCC NGHGI (2019) submissions split per sector.*

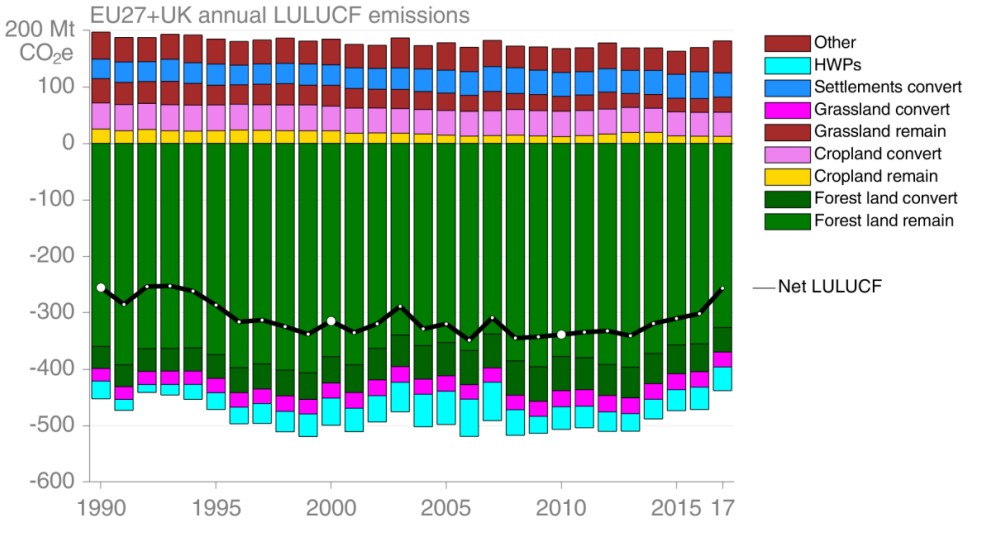

*Figure B1b: EU27+UK total annual GHG emissions from the LULUCF sector split in clases and sub-*
*classes.*



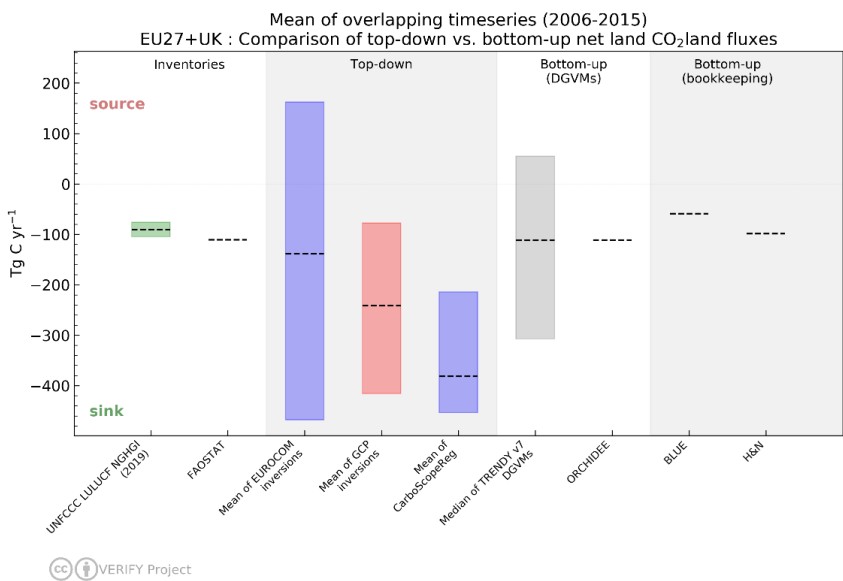

*Figure B1c: Unfolded overlapping (2006-2015) mean column from Figure 10, section 3.3.4. The dotted lines represent the mean of the time series. The NGHGI UNFCCC uncertainty is calculated for 2018 as the relative error on the NGHGI value, computed with the 95 % confidence interval method 16 %. The uncertainty of the three inversions (EUROCOM, CarboScopeReg and GCP) represents the averaged min/max values of the model ensemble estimates.*

## B2: Source specific methodologies: AD, EFs and uncertainties

*Table B2: Source specific activity data (AD), emission factors (EF) and uncertainty methodology for all current VERIFY and non-VERIFY 2019 data product collection.*

| Data sources $CO_2$ emission calculation | AD/Tier | EFs/Tier | Uncertainty assessment method | Emission data availability |
|---|---|---|---|---|
| **UNFCCC NGHGI (2019)** | Country-specific information consistent with the IPCC Guidelines | IPCC guidelines / Country specific information for higher Tiers | IPCC guidelines (https://www.ipcc-nggip.iges.or.jp/public/2006gl/) for calculating the uncertainty of emissions based on the uncertainty of AD and EF, two different approaches: 1. Error propagation, 2. Monte Carlo Simulation | NGHGI official data (CRFs) are found at https://unfccc.int/process-and-meetings/ transparency-and-reporting/ reporting-and-review-under-the-convention/ greenhouse-gas-inventories-annex-i-parties/ submissions/ national-inventory-submissions-2019 (last access: September 2020). |

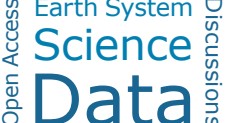

| Fossil $CO_2$ | | | | |
|---|---|---|---|---|
| **BP**<br>**CDIAC**<br>**EIA**<br>**IEA**<br>**GCP**<br>**CEDS** | *For further details, see Andrew (2020)* | | | |
| **EDGAR v5.0** | International Energy Agency (IEA) for fuel combustion<br><br>Food and Agricultural Organisation (FAO) for agriculture<br><br>US Geological Survey (USGS) for industrial processes (e.g. cement, lime, ammonia and ferroalloys)<br><br>GGFR/NOAA for gas flaring<br><br>World Steel Association for iron and steel production<br><br>International Fertilisers Association (IFA) for urea consumption and production<br><br>Complete description of the data sources can be found in Janssens-Maenhout et al. and in Crippa et al. (2019b) | IPCC 2006, Tier 1 or Tier 2 depending on the sector | Tier 1 with error propagation by fuel type for $CO_2$ and accounting for covariances. | https://edgar.jrc.ec.europa.eu/overview.php?v=50_GHG |





| IAP RAS fast-track fCO₂ inversion | Tier3 top-down<br><br>0.1° x 0.1° resolution maps of annual averages of fossil $CO_2$ anthropogenic emissions from EDGAR v4.3.2<br><br>Assimilation of satellite atmospheric concentration data: total column CO from IASI, and tropospheric column $NO_2$ from OMI | Tier3 top-down<br><br>regional inversions of CO and NOx emissions using EMEP/CEIP as prior knowledge of the emissions and CO₂/CO and CO₂/NOx emission ratios associated with the combustion of fossil from EDGARv4.3.2. | Bayesian analysis in the CO and NOx inversions along with propagation of uncertainties in fCO₂/CO and fCO₂/NOx emission ratios | Detailed gridded data can be obtained by contacting the data providers:<br>Gregoire Broquet<br>gregoire.broquet@lsce.ipsl.fr<br>Igor Konovalov<br>konov@ipfran.ru |
|---|---|---|---|---|
| **CO₂ land: bottom-up** | | | | |
| **BLUE** | From LUH2: data on harvest, land cover types (primary, secondary, pasture, crop), and gross land use transitions (e.g. from primary to pasture);Based on Pongratz et al. 2008 and Ramankutty & Foley 1999: Plant functional types (PFTs) of natural vegetation types | Tier 3 (IPCC 2006 guidelines);PFT and land-cover type specific response curves describing the decay and regrowth of vegetation and soil carbon | N/A | Detailed gridded data can be obtained by contacting the data provider:<br>Julia Pongratz:<br>julia.pongratz@geographie.uni-muenchen.de |
| **H&N** | Simple assumptions about C-stock densities (per biome or per biome/country) based on literature | Transient change in C-stocks following a given transition (time dependent EF after an land use transition) | N/A | Detailed gridded data can be obtained by contacting the data provider:<br>Richard Houghton<br>rhoughton@woodwellclimate.org |
| **ECOSSE** | The model is a point model, which provides spatial results by using spatial distributed input data (lateral fluxes are not considered). The model is a TIER 3 approach that is applied on grid map data, polygon organized input data or study sites. | IPCC 2006: Tier 3<br><br>The simulation results will be allocated due to the available information (size of spatial unit, representation of considered land use, etc.). | N/A | Detailed gridded data can be obtained by contacting the data providers:<br>Kuhnert, Matthias<br>matthias.kuhnert@abdn.ac.uk<br>Pete Smith:<br>pete.smith@abdn.ac.uk |



| EPIC-IIASA | Cropland: static 1×1 km cropland mask from CORINE-PELCOM. Initial SOC stock from the Map of organic carbon content in the topsoil (Lugato et al., 2014). "Static" crop management and input intensity by NUTS2 calibrated for 1995-2010 (Balkovič et al., 2013). Crop harvested areas by NUTS2 from EUROSTAT. The model is Tier 3 approach. | IPCC 2006: Tier 3<br><br>Land management and input factors for the cropland remaining cropland category as simulated by the EPIC-IIASA modelling platform, assuming the business-as-usual crop management calibrated for the 1995-2010 period. A 50-ha field is considered in each grid cell. | Sensitivity and uncertainty analysis of EPIC-IIASA regional soil carbon modelling (Balkovič et al. 2020). | Detailed gridded data can be obtained by contacting the data provider: Balcovič Juraj balkovic@iiasa.ac.at |
|---|---|---|---|---|
| **ORCHIDEE** | For the land cover/land use input maps: data on wood harvest from the FAO | Tier 3 model, process based. Any emission factors enter in the form of generic parameters for a given ecosystem type fit against observational data (both site-level and remotely sensed). | None, though some information on uncertainty due to model structure is given by looking at the spread from the TRENDY suite of models, of which ORCHIDEE is a member. | Detailed gridded data can be obtained by contacting the data providers: Matthew Mcgrath matthew.mcgrath@lsce.ipsl.fr Philippe Peylin peylin@lsce.ipsl.fr |
| **TRENDY v7** | For the land cover/land use input maps: data on wood harvest and agricultural land from the FAO | Tier 3 models, process based. Any emission factors enter in the form of generic parameters for a given ecosystem type fit against observational data (both site-level and remotely sensed). | The spread of the 14 TRENDY models used gives an idea of the uncertainty due to model structure in dynamic global vegetation models, as the forcing data was harmonized for all models. | Detailed gridded data can be obtained by contacting the data provider: Sitch, Stephen S.A.Sitch@exeter.ac.uk |
| **Statistical prediction model for $CO_2$ in inland waters** | Hydrosheds 15s (Lehner et al., 2008) and Hydro1K (USGS, 2000) for river network, HYDROLAKES for lakes and reservoirs network and surface area (Messager et al., 2016); | N/A | Monte Carlo runs (uncertainty on $pCO_2$ and gas transfer velocity) | Detailed gridded data can be obtained by contacting the data providers: Ronny Lauerwald Ronny.Lauerwald@ulb.ac.be |





| | | | |
|---|---|---|---|
| | river pCO2 data from GloRiCh (Hartmann et al., 2014), lake pCO$_2$ database from Sobek et al. (2005); river channel slope and width calculated from GLOBE-DEM (GLOBE-Task-Team et al., 1999) and runoff data from Fekete et al. 2002. Geodata for predictors of pCO$_2$ and gas transfer coefficient include air temperature, precipitation and wind speed (Hijmans et al., 2005), population density (CIESIN and CIAT), catchment slope gradient (Hydrosheds 15s), and terrestrial NPP (Zhao et al., 2005) | | | Pierre Regnier Pierre.Regnier@ulb.ac.be |
| **CBM** | national forest inventory data, Tier 2 | EFs directly calculated by model, based on specific parameters (i.e., turnover and decay rates) defined by the user | N/A used from IPCC | Detailed gridded datacan be obtained by contacting the data providers: Roberto Pilli roberto.pilli713@gmail.com Giacomo Grassi Giacomo.GRASSI@ec.europa.eu |
| **EFISCEN** | national forest inventory data, Tier 3 | emission factor is calculated from net balance of growth minus harvest | Sensitivity analysis on EFISCEN V3 in Schelhaas et al. 2007. (the manual) . Total sensitivity is caused by esp. young forest growth, width of volume classes, age of felling and few more. Scenario uncertainty comes on top of this when projecting in future. | Detailed gridded data can be obtained by contacting the data providers: Gert-Jan Nabuurs gert-jan.nabuurs@wur.nl Mart-Jan Schelhaas martjan.schelhaas@wur.nl |
| **FAOSTAT** | FAOSTAT Land Use Domain; Harmonized world soil; ESA CCI; MODIS 6 Burned area products | IPCC guidelines | IPCC (2006, Vol.4, p.10.33) - confidential Uncertainties in estimates of GHG emissions are due to uncertainties in emission factors and activity data. They may be related to, inter alia, natural variability, partitioning fractions, lack of spatial or temporal | Agriculture total and subdomain specific GHG emissions are found for download at http://www.fao.org/faostat/en/#data/GT (last access: June 2020). |



| | | | | |
|---|---|---|---|---|
| | | | coverage, or spatial aggregation. | |
| **CO₂ land: Top-down** | | | | |
| **CarboScopeReg**<br><br>**GCP ensemble** (CTE, CAMS, CarboScope)<br><br>**EUROCOM** (PYVAR-CHIMERE, LUMIA, FLEXINVERT, CarboScopeReg, CTE-Europe) | Tier 3 top-down approach, prior information from fossil emissions, ocean fluxes, and biosphere-atmosphere exchange Spatial resolutions ranging from 1°x1° for certain regions to 4°x5°. EUROCOM uses more than 30 atmospheric stations. CarboScopeReg uses four different settings (as described in Appendix A2) | Tier3 top-down Inversion systems based on atmospheric transport models | CarboScopeReg - Gaussian probability distribution function, where the error covariance matrix includes errors in prior fluxes, observations and transport model representations. GCP: the different methodologies, the land-use and land-cover data set, and the different processes represented trigger the uncertainties between models. a semi-quantitative measure of uncertainty for annual and decadal emissions as best value judgment = at least a 68 % chance (±1σ) EUROCOM: account for source of uncertainties via prior and model and observation error covariance matrices; assessment of the resulting uncertainties in fluxes based on spread | Detailed gridded data can be obtained by contacting the data providers:<br>**CarboScopeReg**: Christoph Gerbig cgerbig@bgc-jena.mpg.de Saqr Munassar smunas@bgc-jena.mpg.de<br><br>**GCP ensembles**: Pierre Friedlingstein P.Friedlingstein@exeter.ac.uk<br><br>**EUROCOM**: Marko Scholze marko.scholze@nateko.lu.se Gregoire Broquet gregoire.broquet@lsce.ipsl.f |

**Data availability**

All raw data files reported in this work which were used for calculations and figures are available for public download at https://doi.org/10.5281/zenodo.4288883 (Petrescu et al., 2020). The data we submitted are reachable with
one click (without the need for entering login and password), with a second click to download the data, consistent with the two click access principle for data published in ESSD (Carlson and Oda, 2018). The data and the DOI number are subject to future updates and only refers to this version of the paper.

**Author contributions**

A. M. R. P., M. J. M.. and A. J. D. designed research and led the discussions; A.M.R.P. wrote the initial draft of the paper and edited all the following versions together with M. J. M., P. P and A. J. D.; M. J. M and R. M. A. made the CO₂ fossil figures and CO₂ land figures respectively; M. J. M and P. P. processed the original data submitted to the VERIFY portal; M. J. M., P. P. and P. B. designed and are managing the web portal; G. P. provided the figures B1a, B1b and B1c and made a very detailed review on a previous version; A. M. R. P. processed the UNFCCC data and
uncertainties; C.Q. helped with making of Figure 4; H. H. A. van der G. was leading the initial discussions within the fossil CO₂ working group and gave valuable suggestions to the manuscript structure; P.C., G. B., F. N. T. C. G., J. P.,



J.-M. G., G. G., G. J. N., P. R., R. L, M. K., J. B., R. P., I. B. K., L. P., P. S., R. L. T., G. C. and A. J. D. read, gave comments and advice on previous versions of the manuscript; all co-authors commented on specific parts related to their data sets; M. J. M, P. P., P. B., F. N. T., P. R., R. L., M. K., J. B., R. P., I. B. K., R. H., M. C., R. G., I. L., C. G., S. M., G. C., G. M., M. S. are data providers.

**Competing interests.**
The authors declare that they have no conflict of interest.

**Acknowledgements**
FAOSTAT statistics are produced and disseminated with the support of its member countries to the FAO regular budget. The views expressed in this publication are those of the author(s) and do not necessarily reflect the views or policies of FAO. Philippe Ciais acknowledges the support of European Research Council Synergy project SyG-2013-610028 IMBALANCE-P and from the ANR CLand Convergence Institute. We acknowledge the work of the entire EDGAR group (Marilena Muntean, Diego Guizzardi, Edwin Schaaf, Jos Olivier). We acknowledge Stephen Sitch and the authors of the DGVMs TRENDY v7 ensemble models for providing us with the data.

**Financial support**
This research has been supported by the European Commission, Horizon 2020 Framework Programme (VERIFY, grant no. 776810).

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
