# Peer review of "The consolidated European synthesis of CO2 emissions and removals for EU27 and UK: 1990-2018"

_Earth System Science Data, 2020_

## Referee Comment (RC1) · Anonymous Referee #1 · 1 Feb 2021

This paper provides a nice synthesis of the current data sets available for CO2 emissions and sinks over EU27 and UK region, with some sector disaggregation. Generally, there is a good agreement between different datasets in terms of magnitude and trend for the fossil fuel emissions, although the only fossil fuel inversion product presented has very large uncertainty (much larger than the other bottom-up datasets). All datasets reporting the sinks from the LULUCF sector have a relatively large uncertainty, and there is not always agreement on the sign between the different approaches, particularly for crop land. This is because of differences in the representation of the processes affecting the direct and indirect emissions from LULUCF, the input data used, as well as the definitions of LULUCF (e.g. managed and unmanaged land and land use

conversions). The paper highlights the complexity of comparing LULUCF data based on different types of approaches and the high uncertainty associated with observation-based inversion approaches. There are several reasons for this, including sparse observing network and high uncertainties in prior and transport models (including effects of boundary conditions when using regional models). The spatial and temporal resolution is another factor that varies a lot from dataset to dataset, as well as the period covered, contributing further to the difficulty of reconciling the different information on the CO2 emissions at regional scale. Finally, the paper provides recommendations to include missing processes (e.g. lateral fluxes between regions) and to reduce the uncertainties associated with different approaches in order to facilitate the integration of all the information available to support the global stock take exercise set out in the Paris Agreement.

The paper is well written and provides a comprehensive source of information regarding a wide range of public datasets, documenting the pros and cons of each data source. I recommend the paper to be published as it is, with only minor corrections (see list of minor comments below).

Minor comments: -Line 62: Replace "CO2 land sources/sinks" with "biogenic CO2 land sources/sinks".

-Line 93: "represent the sum of the effects of sources and sinks".

-Line 98: UK does not use atmospheric observations to complement CO2 (due to difficulty in representing the biogenic fluxes).

-Line 166: Include description of acronyms.

-Lines 169-170: Parenthesis do not match.

-Line 177: Replace "show" with "shown".

-Line 215: Isn't the term "CO2 land fluxes" too generic since the target is LULUCF?

-Line 238: Replace "then" with "than".

-Line 243: Replace "differing" with "differ".

-Table 2: Why is there no contact/lab for BU H&N bookkeeping model?

-Lines 256-257: Numbers with and without LULUCF are not consistent with LULUCF contribution.

-Line 541: Replace "variation trend" with "variation".

-Line 567: "The sink in ORCHIDEE is due to..."

-Line 605-606: "for instance the CO2 fertilization effects..."

-Line 673: "by subtracting from the inversion estimates the emissions..."

-Line 718: It is not clear what are the indirect fluxes on managed land included in NGHGs.

---

## Referee Comment (RC2) · Anonymous Referee #2 · 11 Mar 2021

As described in the title, this manuscript provides a comprehensive overview and summary of the various estimates of CO2 sources and sinks in Europe over the last several decades. This synthesis includes estimates from inventory data, process-based models and inverse models and are further sub-divided into those from fossil fuel emissions and land-based emissions, as well as broken down by sector/industry as well as by land-use types. The paper devotes most of its attention to land-based CO2 fluxes, owing to their complexity and large uncertainty. The various datasets presented in the manuscript often show some significant differences and the authors have done a good job providing explanations for the most likely underlying reasons for these discrepencies. In fact, this is a real strength of this manuscript, highlighting all the many ways

in which data that attempts to measure and quantify the same specific CO2 fluxes can vary due to differences in definitions, which underlying processes are included, and how those processes are represented. The authors provide useful suggestions for further reconciling and harmonizing the multiple datasets. The paper is well written and organized. I recommend it for publication, subject to some minor changes and suggestions, as highlighted below.

1) The Reference section contains three different references for "Petrescu et al. 2020" (which are also cited in the text). To avoid confusion, it would be helpful to relabel these as "Petrescu et al. 2020a" etc.

2) The LUH2 dataset is cited with "Hurtt et al. 2011", but that paper refers to the LUH1 dataset. LUH2 should be cited with Hurtt et al. 2020 (which is already included in the Reference list), or potentially with both Hurtt et al. 2011 and 2020.

3) Line 432: change "emission" to "emissions"

4) Line 431: change "taking into account of the" to "taking into account the"

5) Line 457: change "except a daily" to "except for a daily"

6) Line 461: change "extend" to "extent"

7) The section around line 490 describes how the ESACCI-LUH2v2 dataset assumes that shrublands are equivalent to forest. A rough estimate of the impact of this assumption for the representation of forest area in Europe is included. However, it seems like this could be explicitly quantified from the data, rather than just estimated – is that possible?

8) The description around line 496 about the forest area data from FAOSTAT could use some additional clarification. If FAOSTAT provides the current forest area, not just the FL-FL category, then would it not consider both afforestation and deforestation? Also, line 497 states "This area is based on the same land use/land cover maps", but it is not clear to me what these maps are the same as.

[Figure]

9) Line 657: change "emission" to "emissions"
* * *

---

## Author Comment (AC1) · 22 Mar 2021

Dear Topical Editor Nellie Elguindi, Dear Referees and Editorial Board of ESSD,

As requested, we are submitting responses to the referees' comments. We will provide as well a track-change version of the manuscript. We will not refer here to grammar or language corrections, but they will appear in the marked-up manuscript. The lines in the following answers refer to the track-change version of the manuscript. Given that both referees for the companion paper "The consolidated European synthesis of CH4 and N2O emissions for EU27 and UK: 1990–2017" asked us to

merge all data figures in one spreadsheet "data_figures_CO2.xlsx", we did the same for this synthesis and we updated the Zenodo DOI repository with v2 found here: https://doi.org/10.5281/zenodo.4626578
REPLY TO THE REFEREE #1 The authors thank very much Referee #1 for the very positive and thoughtful comments and for the fact that the Referee acknowledges the manuscript as being a comprehensive source of information regarding a wide range of public products, very useful for modelers and the whole scientific community and for quantifying the progresses towards mitigation target assessed through the global stocktake. Below we provide answers to the minor comments posted by Referee #1.

Response to minor comments and changes in manuscript:

Line 62: Replace "$CO_2$ land sources/sinks" with "biogenic $CO_2$ land sources/sinks".

On L162 we define the two $CO_2$ components analyzed in this study as $CO_2$ fossil and $CO_2$ land. After much discussion in the preparation of this manuscript, we choose to follow the general IPCC GPG, which defines "land" in footnote 4: "The IPCC Good Practice Guidance (GPG) for Land Use, Land Use Change and Forestry (IPCC 2003) describes a uniform structure for reporting emissions and removals of greenhouse gases. This format for reporting can be seen as "land based"; all land in the country must be identified as having remained in one of six classes since a previous survey, or as having changed to a different (identified) class in that period. According to IPCC SRCCL: Land covers the terrestrial portion of the biosphere that comprises the natural resources (soil, near surface air, vegetation and other biota, and water) the ecological processes, topography, and human settlements and infrastructure that operate within that system". Some communities prefer "biogenic" to describe these fluxes, while others found this confusing as fluxes from unmanaged forests, for example, are "biogenic" but not included in inventories reported to the UNFCCC. As this comparison is central

to our work, we decided that "land" as defined by the IPCC was a good compromise. We added this explanation to footnote 4.

Line 93: "represent the sum of the effects of sources and sinks". We made the correction.

Line 98: UK does not use atmospheric observations to complement CO2 (due to difficulty in representing the biogenic fluxes). The referee is right, the UK uses inverse observations only for CH4 emissions and not for CO2. However, this introduction paragraph (L87-L99) refers in general to GHGs.

Line 166: Include description of acronyms. We added acronyms for EDGAR, FAOSTAT, BP, CDIAC, EIA and IEA. GCP is explained on L124.

Lines 169-170: Parenthesis do not match. We made the correction.

Line 177: Replace "show" with "shown". We made the correction (now on L179).

Line 215: Isn't the term "CO2 land fluxes" too generic since the target is LULUCF? Indeed the target is the LULUCF sector and its component classes: forest, cropland, grassland, wetlands, settlements, other land and harvest. We decided to use "land" fluxes according to the UNFCCC definition (footnote 4): "The IPCC Good Practice Guidance (GPG) for Land Use, Land Use Change and Forestry (IPCC 2003) describes a uniform structure for reporting emissions and removals of greenhouse gases. This format for reporting can be seen as "land based"; all land in the country must be identified as having remained in one of six classes since a previous survey, or as having changed to a different (identified) class in that period. According to IPCC SRCCL: Land covers the terrestrial portion of the biosphere that comprises the natural resources (soil, near surface air, vegetation and other biota, and water) the ecological processes, topography, and human settlements and infrastructure that operate within that system".

Line 238: Replace "then" with "than". We made the correction on L243.

Line 243: Replace "differing" with "differ". We made the correction.

Table 2: Why is there no contact/lab for BU H&N bookkeeping model? We added the Woodwell Climate Research Center.

Lines 256-257: Numbers with and without LULUCF are not consistent with LULUCF contribution. This is because the numbers for EU27+UK with and without LULUCF are in CO2eq and include contribution of CH4 and N2O as well. The number we report for LULUCF only (0.28 Gt CO2) is only for CO2.

Line 541: Replace "variation trend" with "variation". We made the suggested correction the new L552.

Line 567: "The sink in ORCHIDEE is due to. . ." We included "to" on the new L579.

Line 605-606: "for instance the CO2 fertilization effects. . ." We deleted "by" on the new L617.

Line 673: "by subtracting from the inversion estimates the emissions. . ." We deleted "of" on the new L684.

Line 718: It is not clear what are the indirect fluxes on managed land included in NGHGs. According to IPCC (2010), land fluxes can be differentiated into three processes: (1) direct anthropogenic effects (land use and land use change, e.g., harvest, other management, deforestation), (2) indirect anthropogenic effects (e.g., changes induced by human-induced climate change, including CO2 fertilization and nitrogen deposition changes), and (3) natural effects (i.e., that would happen without human-caused climate change, such as natural disturbances). The UNFCCC NGHG inventories use the notion of managed land as a proxy for anthropogenic emissions (IPCC, 2006) and hence in practice include most or all (depending on the specific method) indirect emissions into their anthropogenic estimates (Petrescu et al., 2020b). We added on L729 the following explanation: "(indirect fluxes on managed land included in NGHGIs and FAOSTAT e.g., changes due to human-induced climate change, including CO2 fertilization and nitrogen deposition changes) (Petrescu et al., 2020b)".

---

## Author Comment (AC2) · 22 Mar 2021

Dear Topical Editor Nellie Elguindi, Dear Referees and Editorial Board of ESSD,

As requested, we are submitting responses to the referees' comments. We will provide as well a track-change version of the manuscript. We will not refer here to grammar or language corrections, but they will appear in the marked-up manuscript. The lines in the following answers refer to the track-change version of the manuscript. Given that both referees for the companion paper "The consolidated European synthesis of CH4 and N2O emissions for EU27 and UK: 1990–2017" asked us to

merge all data figures in one spreadsheet "data_figures_CO2.xlsx", we did the same for this synthesis and we updated the Zenodo DOI repository with v2 found here: https://doi.org/10.5281/zenodo.4626578
REPLY TO THE REFEREE #2 The authors thank Referee #2 for acknowledging this study as being a comprehensive overview and summary of the various estimates of $CO_2$ sources and sinks in Europe, well written and well structured. We thank Referee #2 for the comments to which we answer below.

1) The Reference section contains three different references for "Petrescu et al. 2020" (which are also cited in the text). To avoid confusion, it would be helpful to relabel these as "Petrescu et al. 2020a" etc. We agree and labeled the three Petrescu references as following: a) Zenodo data sets; b) ESSD AFOLU paper, and the companion synthesis paper on CH4 and N2O with 2021, in press.

2) The LUH2 dataset is cited with "Hurtt et al. 2011", but that paper refers to the LUH1 dataset. LUH2 should be cited with Hurtt et al. 2020 (which is already included in the Reference list), or potentially with both Hurtt et al. 2011 and 2020. Thank you. We changed everywhere to Hurtt et al., 2020 as we refer to LUH2.

3) Line 432: change "emission" to "emissions" We made the correction.

4) Line 431: change "taking into account of the" to "taking into account the" We deleted "of".

5) Line 457: change "except a daily" to "except for a daily" We added "for".

6) Line 461: change "extend" to "extent" We made the correction.

7) The section around line 490 describes how the ESACCI-LUH2v2 dataset assumes that shrublands are equivalent to forest. A rough estimate of the impact of this assumption for the representation of forest area in Europe is included. However, it seems like this could be explicitly quantified from the data, rather than just estimated – is that possible?

The reviewer would like to see a more precise calculation of the impact of classifying all shrubs as forests in the ESACCI-LUH2v2 product on the NBP fluxes of the EU-27+UK. Note first that ESACCI product effectively distinguishes shrub from forest but that we have further grouped them given that the ORCHIDEE model does not distinguish shrubs. While possible, the scientific benefit is unclear. As mentioned in the paper, the definition of "forest land" varies by Member State in the UNFCCC reporting (table 6.10 in the EU NIR for 2020 gives a nice summary of the forest question https://unfccc.int/documents/228021). Changing all shrub land to some other land type in the simulations (either grassland or cropland) would likely get us closer to the "real" answer, but it's not clear how close, or if we would over- or under-estimate it, given that the definition of forest used in the ESA-CCI maps does not match all of the individual definitions for each member state; one definition of "forest" is applied across the whole ESA-CCI product. Due to the lack of a clear mapping between the ESA-CCI land cover classes and the Member State definitions for land cover and land use, we are not convinced that a more work-intensive estimation, while perhaps more precise, would be demonstratively more accurate than the estimation we already give. However, we have indeed carried out more precise estimate, looking at the amount of land area classified as "shrubs" and "tree" in the original ESA-CCI land cover classification from 2015 for the EU-27+UK. We find a total of 1.01 Mkm2 for trees and 0.498 Mkm2 for shrubs, which means the proportion of shrubs is significantly higher (50 %) than our previous estimate (10 %). A similar analysis for the FAOSTAT domain Land Cover, which maps and disseminates the areas of MODIS and ESA-CCI land cover classes to the SEEA land cover categories http://www.fao.org/faostat/en/#data/LC, shows that shrub-covered areas are around 20 % of that of forested areas. Given the uncertainty in land cover definitions mentioned above, we have modified the text in the manuscript as following, L490-499: "For this study, the ORCHIDEE model used a socalled ESA-CCI LUH2v2 PFT distribution (a combination of the ESA-CCI land cover map for 2015 with the historical land cover reconstruction from LUH2 (Lurton et al., 2020)), and assumes that the shrub land cover classes are equivalent to forest. In terms of area, the original ESA-CCI product corresponding to our domain of the EU-27+UK shows shrub land equal to about 50 % of the tree area in 2015. A similar analysis using the FAOSTAT domain Land Cover, which maps and disseminates the areas of MODIS and ESA-CCI land cover classes to the SEEA land cover categories (http://www.fao.org/faostat/en/#data/LC), shows that shrub-covered areas are around 20 % of that of forested areas for the EU-27+UK. The impact of classifying shrubs as "forests" on the total carbon fluxes could therefore account for a significant percentage of the differences between ORCHIDEE and other results in Figure 6."

8) The description around line 496 about the forest area data from FAOSTAT could use some additional clarification. If FAOSTAT provides the current forest area, not just the FL-FL category, then would it not consider both afforestation and deforestation? Also, line 497 states "This area is based on the same land use/land cover maps", but it is not clear to me what these maps are the same as. We agree that this is an unclear explanation of the area used by FAOSTAT and we corrected as following L507-509: "FAOSTAT forest land area is based on country statistics from the FAO/FRA process and includes not only forest remaining forest area but all forested land, including af-forestation."

9) Line 657: change "emission" to "emissions" We corrected, now on L668.

References: IPCC: Good Practice Guidance for Land use, Land use Change and Forestry, available at: https://www.ipcc-nggip.iges.or.jp/public/ gpglu-lucf/gpglulucf_files/GPG_LULUCF_FULL.pdf (last access: January 2020), 2003.

IPCC: Uncertainties, chap. 3, in: 2006 IPCC Guidelines for National Greenhouse Gas Inventories, available at: https://www.ipcc-nggip.iges.or.jp/public/2006gl/pdf/1_Volume1/V1_3_Ch3_Uncertainties.pdf (last access: December 2019), 2006.

IPCC: Revisiting the Use of Managed Land as a Proxy for Estimating National Anthropogenic Emissions and Removals, edited by: Eggleston, H. S., Srivastava, N., Tanabe, K., and Baasansuren, J., Meeting Report, INPE, Sao José dos Campos, Brazil, 5–7 May 2009, Pub. IGES, Japan, 2010.

Petrescu, A. M. R., Peters, G. P., Janssens-Maenhout, G., Ciais, P., Tubiello, F. N., Grassi, G., Nabuurs, G.-J., Leip, A., Carmona-Garcia, G., Winiwarter, W., Höglund-Isaksson, L., Günther, D., Solazzo, E., Kiesow, A., Bastos, A., Pongratz, J., Nabel, J. E. M. S., Conchedda, G., Pilli, R., Andrew, R. M., Schelhaas, M.-J., and Dolman, A. J.: European anthropogenic AFOLU greenhouse gas emissions: a review and benchmark data, Earth Syst. Sci. Data, 12, 961–1001, https://doi.org/10.5194/essd-12-961-2020, 2020b.

Hurtt, G. C., Chini, L., Sahajpal, R., Frolking, S., Bodirsky, B. L., Calvin, K., Doelman, J. C., Fisk, J., Fujimori, S., Klein Goldewijk, K., Hasegawa, T., Havlik, P., Heinimann, A., Humpenöder, F., Jungclaus, J., Kaplan, J. O., Kennedy, J., Krisztin, T., Lawrence, D., Lawrence, P., Ma, L., Mertz, O., Pongratz, J., Popp, A., Poulter, B., Riahi, K., Shevliakova, E., Stehfest, E., Thornton, P., Tubiello, F. N., van Vuuren, D. P., and Zhang, X.: Harmonization of global land use change and management for the period 850–2100 (LUH2) for CMIP6, Geosci. Model Dev., 13, 5425–5464, https://doi.org/10.5194/gmd-13-5425-2020, 2020.

Additional changes :

We added A. to co-author Richard Houghton L480: Caption Figure 6: we corrected the mean common period to 2006-2015.